# The FlgN chaperone activates the Na$^+$-driven engine of the *Salmonella* flagellar protein export apparatus

Tohru Minamino [1✉], Miki Kinoshita [1], Yusuke V. Morimoto [2,3] & Keiichi Namba [1,4,5]

The bacterial flagellar protein export machinery consists of a transmembrane export gate complex and a cytoplasmic ATPase complex. The gate complex has two intrinsic and distinct H$^+$-driven and Na$^+$-driven engines to drive the export of flagellar structural proteins. *Salmonella* wild-type cells preferentially use the H$^+$-driven engine under a variety of environmental conditions. To address how the Na$^+$-driven engine is activated, we analyzed the *fliJ(Δ13–24) fliH(Δ96–97)* mutant and found that the interaction of the FlgN chaperone with FlhA activates the Na$^+$-driven engine when the ATPase complex becomes non-functional. A similar activation can be observed with either of two single-residue substitutions in FlhA. Thus, it is likely that the FlgN-FlhA interaction generates a conformational change in FlhA that allows it to function as a Na$^+$ channel. We propose that this type of activation would be useful for flagellar construction under conditions in which the proton motive force is severely restricted.

[1] Graduate School of Frontier Biosciences, Osaka University, Osaka, Japan. [2] Department of Physics and Information Technology, Faculty of Computer Science and Systems Engineering, Kyushu Institute of Technology, Fukuoka, Japan. [3] Japan Science and Technology Agency, PRESTO, Saitama, Japan. [4] RIKEN SPring-8 Center and Center for Biosystems Dynamics Research, Osaka, Japan. [5] JEOL YOKOGUSHI Research Alliance Laboratories, Osaka University, Osaka, Japan. ✉email: tohru@fbs.osaka-u.ac.jp

The bacterial flagellum is a macromolecular protein complex responsible for rapid and efficient movement of bacterial cells towards more suitable environments. The flagellum is composed of the basal body, which acts as a rotary motor, the hook, which serves as a universal joint, and the filament, which forms a helical propeller[1,2]. To construct the flagellum on the cell surface, a specialized protein export machinery located at the flagellar base transports flagellar structural subunits from the cytoplasm to the distal end of the growing flagellar structure. The flagellar export machinery is composed of a transmembrane export gate complex powered by a proton motive force (PMF) across the cytoplasmic membrane and a cytoplasmic ATPase ring complex[3,4]. This export machinery is structurally and functionally similar to virulence-related type III secretion systems of pathogenic bacteria, which directly inject virulence effector proteins into eukaryotic host cells[5].

The transmembrane export gate complex is located inside the basal body MS ring and acts as a proton/protein antiporter to drive $H^+$-coupled protein translocation across the cytoplasmic membrane[3,4]. FliP, FliQ, and FliR form a polypeptide channel complex for the translocation of export substrates across the cytoplasmic membrane[6,7]. FlhB associates with the FliP/FliQ/FliR complex and is postulated to coordinate opening of the polypeptide channel[8]. FlhA associates not only with the FliP/FliQ/FliR complex but also with the MS ring[9]. Because FlhA promotes the transit of both $H^+$ and $Na^+$ across the cytoplasmic membrane, it seems to act as the export engine of the export gate complex[10,11]. The C-terminal cytoplasmic domains of FlhA ($FlhA_C$) and FlhB ($FlhB_C$) project into the central cavity of the basal body C ring and form a docking platform for the cytoplasmic ATPase complex (FliH, FliI, FliJ), flagellar chaperones (FlgN, FliS, FliT) and export substrates[12,13]. This docking platform coordinates the order of flagellar protein export with assembly in a highly organized and well-controlled manner[14].

FliH, FliI, and FliJ form the cytoplasmic ATPase ring complex at the flagellar base[15]. This structure is not essential for flagellar protein export in Salmonella enterica serovar Typhimurium (hereafter referred to as Salmonella)[16–18], but it ensures robust and efficient coupling of energy to flagellar protein export[19,20]. The FliI ATPase

hydrolyses ATP and activates the export gate complex through an interaction between $FlhA_C$ and FliJ, which is located at the center of the FliI hexamer ring, thereby allowing the transmembrane export gate complex to become active in coupling influx of $H^+$ through the FlhA ion channel with the translocation of export substrates through the FliP/FliQ/FliR polypeptide channel[21–23].

The export gate complex can also use a sodium motive force (SMF) across the cytoplasmic membrane to drive $Na^+$-coupled protein export when the cytoplasmic ATPase ring complex is absent or nonfunctional[10]. Because the FlhA ion channel conducts both $H^+$ and $Na^+$, it is plausible that the ATPase complex may switch the ion channel properties of FlhA from a dual ion mode to a highly efficient $H^+$ channel through an interaction between $FlhA_C$ and FliJ[10]. However, it remains unknown how this might happen.

In planktonic Salmonella wild-type cells, the transmembrane export gate complex preferentially uses the PMF to transport flagellar structural subunits to the cell exterior under a variety of environmental conditions[16–18,21]. However, when the cytoplasmic ATPase ring complex becomes nonfunctional, as during biofilm formation[24], the export gate preferentially uses the SMF over a wide range of external pH[10]. A subpopulation of planktonic cells can rapidly move in the biofilm structure by rotating flagella to keep cells in the biofilm alive and healthy[25]. The second messenger molecule 3′–5′ cyclic diguanylate monophosphate, which induces biofilm formation, not only inhibits the transcription of flagellar genes but also binds to the FliI ATPase to suppress flagellar assembly of the cells in the biofilm[24]. The total PMF seems to be quite low in the cells living in the biofilm because the membrane voltage is quite small[26]. These observations suggest that the flagellar protein export apparatus would evolve to retain the $Na^+$-driven export engine so that flagellated cells could arise in biofilms.

To clarify how the $Na^+$-driven export engine is activated, we analyzed the export properties of the Salmonella MM104H-3 [fliJ(Δ13–24) fliH(Δ96–97)] strain (hereafter referred to as $J_{(Δ13–24)} H^*$, Table 1), in which the extragenic fliH suppressor

**Table 1 Summary for important properties of Salmonella strains with respect to motility, flagellar protein export and flagellar assembly.**

| Strains | Abbreviated name | NaCl (mM) | Motility | Flagellar protein export | | Flagellar assembly | |
|---|---|---|---|---|---|---|---|
| | | | | FlgD secretion | FliC secretion | HBB[a] | Filament |
| SJW1103 (Wild-type) | WT | 0 | ++++ | +++ | +++ | N.A.[b] | +++ |
| | | 100 | ++++ | +++ | +++ | + | ++++ |
| MM104-3 [fliJ(Δ13-24) fliH*] | $J_{(Δ13-24)}$ H | 0 | + | +++ | +++ | N.A. | ++ |
| | | 100 | ++ | ++++ | ++++ | + | +++ |
| MMHI1017(ΔfliHI flhB*) | ΔHI B* | 0 | +/− | +/− | − | N.A. | N.A. |
| | | 100 | ++ | +++ | ++ | N.A. | ++ |
| MMHIJ1017 (ΔfliHIJ flhB*) | ΔHIJ B* | 0 | +/− | − | − | N.A. | N.A. |
| | | 100 | + | + | + | N.A. | + |
| MMHIJ1017-2 [ΔfliHIJ flhB* flhA(D456V)] | ΔHIJ B* A1* | 100 | N.A. | +++ | +++ | N.A. | ++ |
| MMHIJ1017 [ΔfliHIJ flhB* flhA(T490M)] | ΔHIJ B* A2* | 100 | N.A. | +++ | +++ | N.A. | ++ |
| MM9001(ΔflgN) | ΔN | 100 | + | +++ | +++++ | + | +/− |
| MM9003 [fliJ(Δ13-24) fliH* ΔflgN] | $J_{(Δ13-24)}$ H ΔN | 100 | − | − | − | − | − |
| MM9003-2 [fliJ(Δ13-24) fliH* ΔflgN flhA(D456V)] | $J_{(Δ13-24)}$ H ΔN A1* | 100 | + | ++++ | ++++ | N.A. | N.A. |
| MM9003-3 [fliJ(Δ13-24) fliH* ΔflgN flhA (T490M)] | $J_{(Δ13-24)}$ H ΔN A2* | 100 | + | ++++ | ++++ | N.A. | N.A. |
| MM9002 (ΔfliHI flhB* ΔflgN) | ΔHI B* ΔN | 100 | − | − | − | N.A. | − |
| MM9004 (ΔfliHIJ flhB* ΔflgN) | ΔHIJ B* ΔN | 100 | − | − | − | N.A. | − |
| MM9004-2 [ΔfliHIJ flhB* ΔflgN flhA(D456V)] | ΔHIJ B* ΔN A1* | 100 | N.A. | +++ | +++ | N.A. | +/− |
| MM9004-3 [ΔfliHIJ flhB* ΔflgN flhA(T490M)] | ΔHIJ B* ΔN A2* | 100 | N.A. | +++ | +++ | N.A. | +/− |

[a]HBB hook-basal body.
[b]N.A. not analyzed.

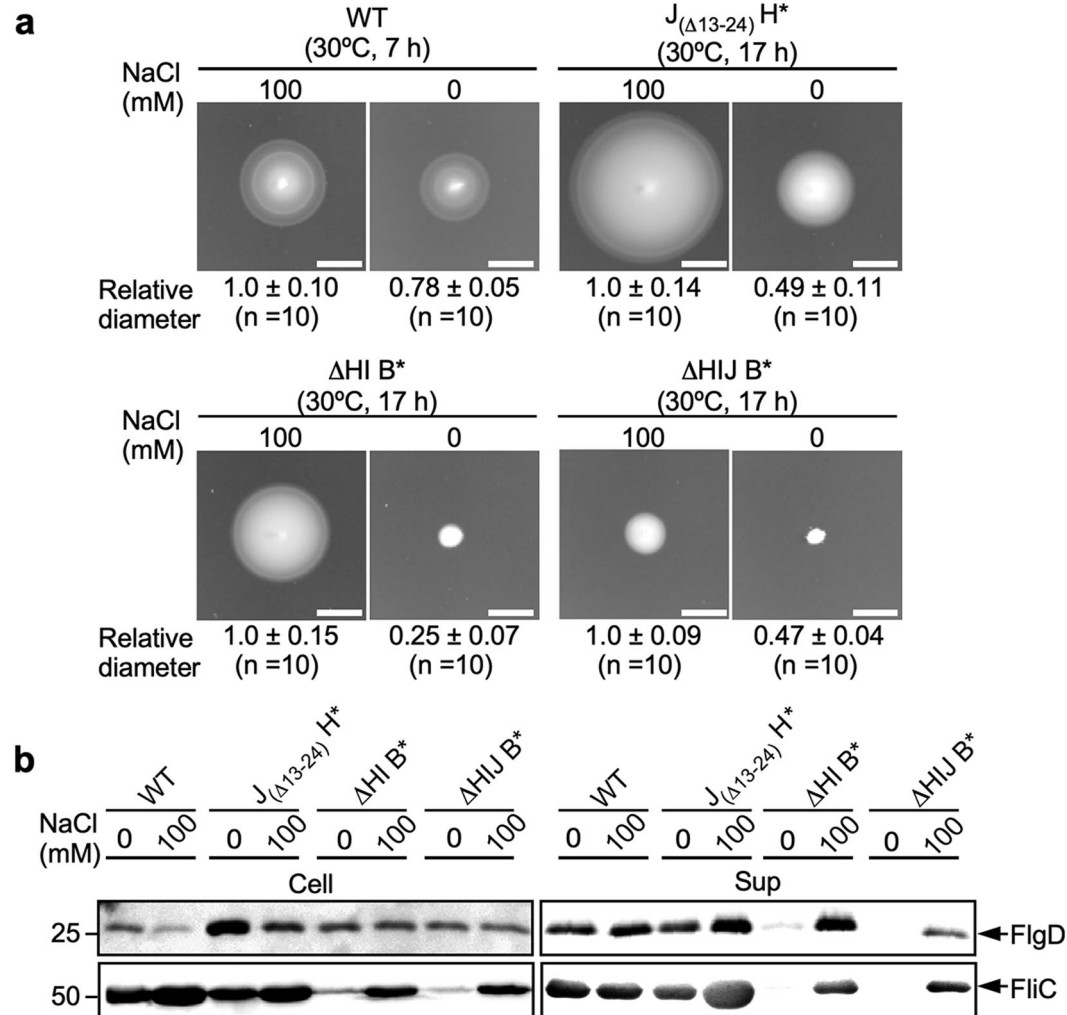

**Fig. 1 Effect of Na$^+$ ions on flagellar protein export. a** Motility of SJW1103 (wild-type, indicated as WT), MM104H-3 [*fliJ(Δ13–24) fliH(Δ96–97)*, indicated as J$_{(Δ13–24)}$ H*], MMHIO117 [*ΔfliH-fliI flhB(P28T)*, indicated as ΔHI B*] and MMHIJO117 [*ΔfliH-fliI-fliJ flhB(P28T)*, indicated as ΔHIJ B*] in 0.35% soft agar plates in the absence and presence of 100 mM NaCl. The diameter of the motility ring of 10 colonies of each strain was measured in the presence and absence of 100 mM NaCl. The average diameter of the motility ring of each strain grown in the presence of 100 mM NaCl was set to 1.0, and then relative diameter of the motility ring of cells grown in the absence of NaCl was calculated (mean ± SD, $n = 10$). Scar bar, 1.0 cm. **b** Effect of Na$^+$ on flagellar protein export at external pH 7.5. Immunoblotting, using polyclonal anti-FlgD or anti-FliC antibody, of whole-cell proteins (Cell) and culture supernatant fractions (Sup) prepared from the above strains grown exponentially at 30 °C in T-broth (pH 7.5) with or without 100 mM NaCl. The regions of interest were cropped from original immunoblots shown in Supplementary Fig. 10.

mutation partially rescues the interaction of FliJ(Δ13–24) with FlhA$_C$, thereby restoring flagellar formation in the presence of the *fliJ(Δ13–24)* mutation[21]. We show that the J$_{(Δ13–24)}$ H* cells use the SMF to produce flagella and that an interaction of FlhA$_C$ with FlgN is essential for this Na$^+$-coupled protein export.

## Results

**Effect of the SMF on flagellar protein export by the J$_{(Δ13–24)}$ H* strain**. To clarify whether an altered interaction of FliJ with FlhA$_C$ induces opening of a Na$^+$ channel in FlhA, we analyzed the effect of the SMF on flagellar formation by the J$_{(Δ13–24)}$ H* strain, in which FliJ has a decreased affinity for FlhA$_C$[21]. We set the external pH at 7.5 to diminish the chemical potential gradient of H$^{+}$ [27] and confirmed that was the case by using a pH indicator protein, pHluorin(M153R)[28,29] to show that the intracellular pH of *Salmonella* cells was 7.41 ± 0.05. The results for all strains we used in this study are qualitatively summarized in Table 1.

Motility of the J$_{(Δ13–24)}$ H* strain was better in the presence of 100 mM NaCl than in its absence (Fig. 1a and Supplementary Fig. 1). The amount of FlgD and FliC secreted by these cells was also higher in the presence of 100 mM NaCl than in its absence (Fig. 1b). Because the growth rate of *Salmonella* cells is slower under no-salt conditions compared to in the presence of 100 mM NaCl or 100 mM KCl (Supplementary Fig. 2), we also analyzed the motility of the J$_{(Δ13–24)}$ H* cells in the presence of 100 mM KCl. These cells were less motile in the presence of 100 mM KCl than in the presence of 100 mM NaCl (Supplementary Fig. 3a). Also, unlike NaCl, KCl did not enhance the secretion level of FliC (Supplementary Fig. 3b). These results demonstrate that Na$^+$ facilitates protein export by the J$_{(Δ13–24)}$ H* cells, whereas neither motility nor flagellar protein export by wild-type cells showed any Na$^+$ dependence (Fig. 1 and Supplementary Fig. 3), in agreement with a previous report[10].

To quantify the efficiency of flagellar assembly, we labelled the filaments with a fluorescent dye (Fig. 2a) and measured the

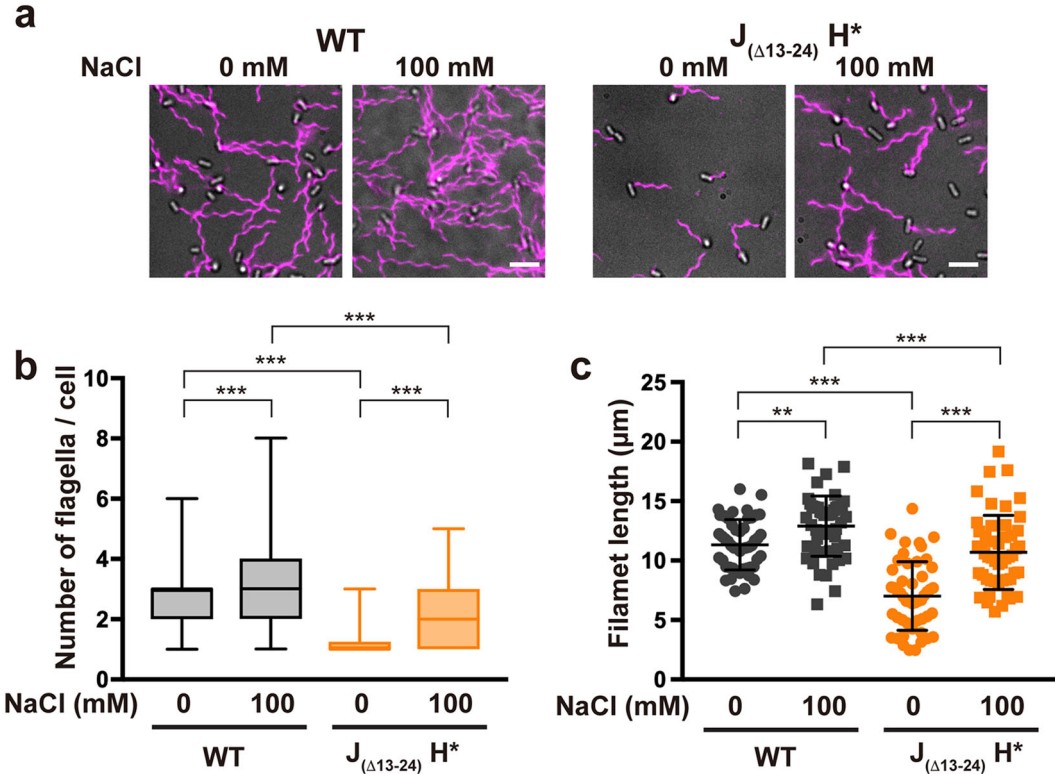

**Fig. 2 Measurements of the number and length of flagellar filaments produced by the J$_{(Δ13-24)}$ H* strain. a** Fluorescent images of the SJW1103 (WT) and MM104H-3 (J$_{(Δ13-24)}$ H*) cells. The cells were grown in T-broth (pH 7.5) with or without 100 mM NaCl until the cells reached the stationary phase, and then flagellar filaments were labelled with a fluorescent dye, Alexa Fluor 594. The fluorescence images of the filaments labelled with Alexa Fluor 594 (magenta) were merged with the bright field images of the cell bodies. Scale bar, 5.0 μm. **b** Average number of flagellar filaments. Box plots show the number of the flagellar filaments in the WT and J$_{(Δ13-24)}$ H* cells. Lower and upper box boundaries are 25th and 75th percentiles, respectively. The line in the middle of the box is median. Lower and upper error lines are the smallest and largest values, respectively. More than 110 cells were counted. **c** Scatter plots of flagellar filament length. Filament length is the average of 50 filaments, and vertical lines are standard deviations. Comparisons between datasets were performed using a two-tailed Student's t-test. A P-value of <0.05 was considered to be statistically significant difference. **P < 0.01; ***P < 0.001. (Also see Supplementary Table 1).

number and length of the filaments (Supplementary Table 1). Wild-type cells produced an average of $2.7 \pm 1.1$ filaments per cell (mean $\pm$ SD, $n = 152$) in the absence of NaCl and $3.3 \pm 1.5$ filaments per cell ($n = 153$) in the presence of 100 mM NaCl (Fig. 2b). The average filament length was $11.3 \pm 2.1$ μm ($n = 50$) in the absence of NaCl and $12.9 \pm 2.5$ μm ($n = 50$) in the presence of 100 mM NaCl (Fig. 2c). Because the transcription levels of flagellar genes were not increased by adding 100 mM NaCl[10], we suggest that the wild-type protein export apparatus may also utilize the SMF to some extent. In the absence of NaCl, 38.5% of the J$_{(Δ13-24)}$ H* cells had no visible filaments. The remaining population produced an average of $1.3 \pm 0.5$ filaments per cell ($n = 118$) (Fig. 2b). The average filament length was $7.0 \pm 2.9$ μm ($n = 50$), which is about 1.6-fold shorter than the length of the wild-type filaments in the absence of NaCl (Fig. 2c), indicating that the growth rate of filaments in these cells is slower than in the wild-type. In contrast, 87.3% of the J$_{(Δ13-24)}$ H* cells produced the filaments in the presence of 100 mM NaCl, with an average number of $2.0 \pm 1.0$ per cell ($n = 145$) (Fig. 2b). The average filament length was $10.7 \pm 3.1$ μm ($n = 50$), which is about 1.5-fold longer than the filament length of cells grown without NaCl (Fig. 2c). These results suggest that the transmembrane export gate complex of this strain uses the SMF in addition to the PMF to transport flagellar structural proteins during flagellar assembly. Based on these results, we propose that an altered interaction between FliJ and FlhA$_C$ activates a Na$^+$-driven export engine to promote Na$^+$-coupled flagellar protein export.

**Effect of FlgN on flagellar protein export by J$_{(Δ13-24)}$ H* cells.** The *Salmonella* MMHI0117 [*ΔfliH-fliI flhB(P28T)*] strain (hereafter referred to as ΔHI B*, Table 1) is a pseudorevertant isolated from a mutant with a deletion of the two genes that form the cytoplasmic ATPase complex[16]. The ΔHI B* cells also preferentially use the SMF rather than the PMF to produce flagella and support some motility (Fig. 1a)[10]. Under no-salt conditions, FliC is not expressed in the ΔHI B* strain (Fig. 1b) because FlgM, the transcription repressor of class 3 flagellar proteins such as FliC until hook assembly is complete[30,31], is also not secreted out of the cytoplasm (Supplementary Fig. 4). In agreement with a previous report[10], addition of 100 mM KCl did not enhance motility or flagellar protein export by the ΔHI B* cells (Supplementary Fig. 3). It has been shown that a nonfunctional variant of FliJ, GST-FliJ, binds to FlhA and inhibits flagellar protein export by the ΔHI B* cells[32]. Therefore, to identify the flagellar protein required for activation of the Na$^+$-driven export engine, GST-FliJ was overexpressed in ΔHI B* cells, and whole-cell lysates were subjected to GST affinity chromatography. In addition to FlhA, FlgN also co-purified with GST-FliJ, but not with GST alone (Supplementary Fig. 5). FlgN is a flagellar export chaperone specific for two hook-filament junction proteins, FlgK and FlgL[33]. The FlgN chaperone protects these two proteins from proteolysis in the cytoplasm[34] and also facilitates the docking of FlgK and FlgL to FlhA$_C$ to expedite rapid and efficient export of these proteins[35–37]. Unlike the FliS and FliT chaperones, which

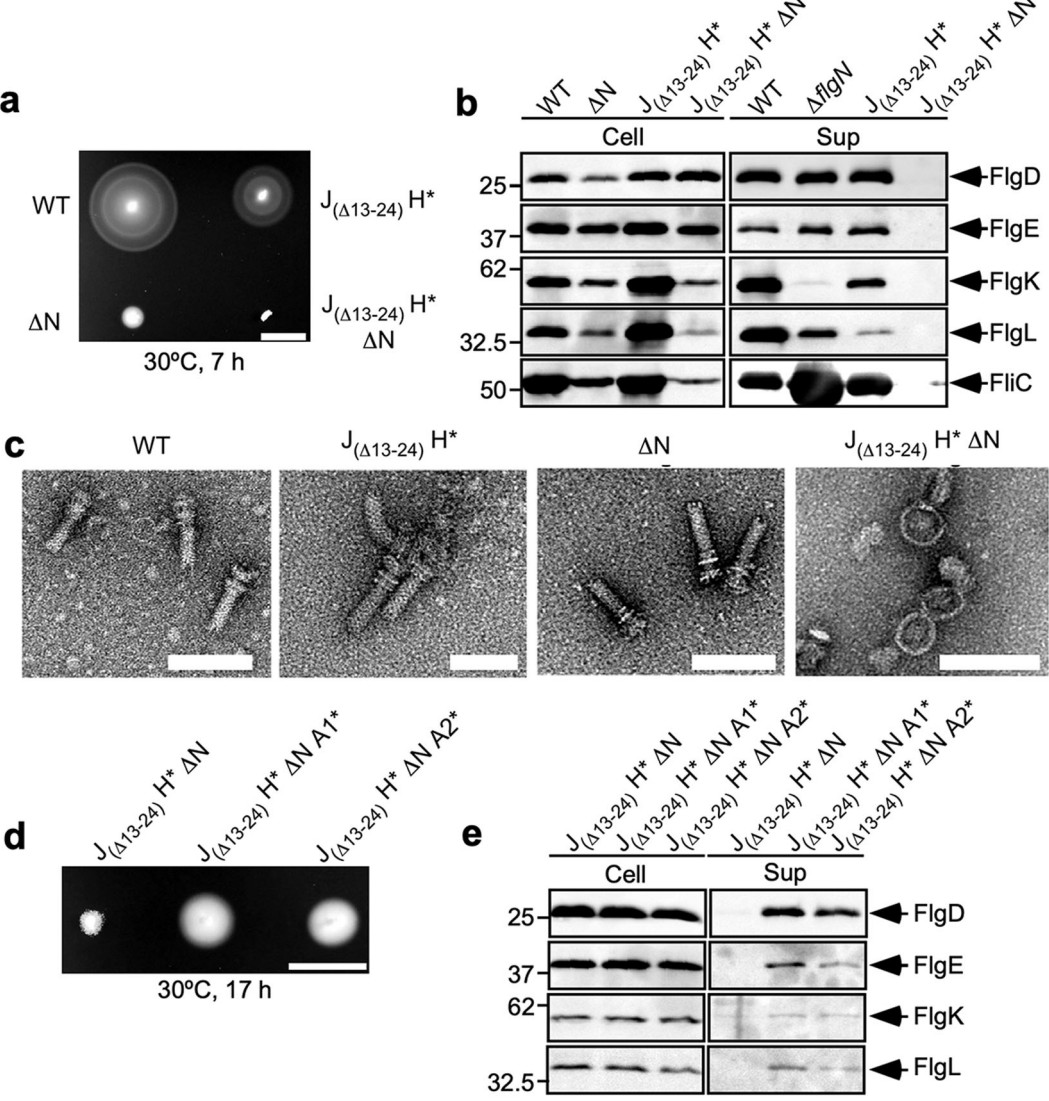

**Fig. 3 Effect of FlgN deletion on flagellar protein export by *fliJ(Δ13–24) fliH* * cells. a** Motility of SJW1103 (WT), MM104H-3 (J$_{(Δ13-24)}$ H*), MM9001 (ΔN) and MM9003 (J$_{(Δ13-24)}$ H* ΔN) in 0.35% soft agar plates containing 100 mM NaCl. Scale bar, 1.0 cm. **b** Immunoblotting, using polyclonal anti-FlgD (1st row), anti-FlgE (2nd row), anti-FlgK (3rd row), anti-FlgL (4th row) or anti-FliC (5th row) antibody, of whole-cell proteins (Cell) and culture supernatant fractions (Sup) prepared from the above strains. The regions of interest were cropped from original immunoblots shown in Supplementary Fig. 11a. **c** Electron micrographs of hook-basal bodies isolated from the above stains. Scale bar, 100 nm. **d** Motility of MM9003, MM9003-2 (J$_{(Δ13-24)}$ H* ΔN A1*) and MM9003-3 (J$_{(Δ13-24)}$ H* ΔN A2*) in 0.35% soft agar plates containing 100 mM NaCl. Scale bar, 1.0 cm. **e** Immunoblotting, using polyclonal anti-FlgD (1st row), anti-FlgE (2nd row), anti-FlgK (3rd row), or anti-FlgL (4th row) antibody, of whole-cell proteins (Cell) and culture supernatant fractions (Sup) prepared from the above strains. The regions of interest were cropped from original immunoblots shown in Supplementary Fig. 11b.

require their cognate export substrates, FliC and FliD, respectively, to bind to FlhA$_C$, FlgN binds to FlhA$_C$ with nanomolar affinity even in the absence of FlgK and FlgL[36]. This property of FlgN raises the question of whether its interaction with FlhA$_C$ activates the Na$^+$-driven export engine.

To answer this question, we introduced a Δ*flgN::tetRA* allele into the wild-type and J$_{(Δ13–24)}$ H* strains by P22-mediated transduction to produce the Δ*flgN* and J$_{(Δ13–24)}$ H* Δ*flgN* strains (hereafter referred to as ΔN and J$_{(Δ13–24)}$ H* ΔN, respectively, Table 1) and analyzed motility of these two transductants in soft agar. About 35.2% of the ΔN cells produced a single flagellar filament (Supplementary Fig. 6a), and this level of flagellar synthesis was sufficient to generate a small motility ring on soft agar plates (Fig. 3a). In contrast, the J$_{(Δ13–24)}$ H* ΔN cells were completely nonmotile (Fig. 3a), and no filaments were seen on these cells (Supplementary Fig. 6a).

To discover why the J$_{(Δ13–24)}$ H* ΔN cells do not produce flagellar filaments, we next analyzed the impact of Δ*flgN* on flagellar protein export. The loss of FlgN considerably reduced the levels of FlgK and FlgL secreted by wild-type cells but had little effect on secretion of the hook-capping protein, FlgD or the hook protein, FlgE (Fig. 3b). The ΔN cells produced visible hook-basal bodies (HBBs) (Fig. 3c), in agreement with a previous report[38]. In contrast, the presence of Δ*flgN::tetRA* inhibited the secretion of FlgD and FlgE by the J$_{(Δ13–24)}$ H* cells (Fig. 3b). Whereas the J$_{(Δ13–24)}$ H* cells produced HBBs, the J$_{(Δ13–24)}$ H* cells containing Δ*flgN::tetRA* produced only the MS-C ring structures (Fig. 3c). Also, the intracellular levels of FliC, FlgK and FlgL were much lower in the absence of FlgN than in its presence in the J$_{(Δ13–24)}$ H* mutant background (Fig. 3b), suggesting that FlgN is also required for FlgM secretion. When FlgN was expressed from a pTrc99A based plasmid in the J$_{(Δ13–24)}$ H* ΔN cells, the motility

was restored to a level comparable to that of the $J_{(\Delta 13-24)}$ H* strain (Supplementary Fig. 6b). These results indicate that FlgN becomes essential for the export of all flagellar structural proteins by the $J_{(\Delta 13-24)}$ H* cells.

The ΔHI B* and MMHIJ0117 [Δ*fliH*-*fliI*-*fliJ flhB(P28T)*] (hereafter referred to as ΔHIJ B*, Table 1) cells showed a clear Na$^+$ dependence on flagellar protein export (Fig.1 and Supplementary Fig. 3). To confirm our observations described above, we also introduced the Δ*flgN*::*tetRA* allele into these two strains and found that a loss of FlgN results in a completely nonmotile phenotype (Fig. 4a and Supplementary Fig. 6b). Also, neither FlgD, FliC nor FlgM were seen in the culture supernatants of these two strains containing Δ*flgN*::*tetRA* (Fig. 4b and Supplementary Figs 7 and 8). Therefore, we conclude that FlgN is essential for Na$^+$-coupled protein export by the transmembrane export gate complex when the functional cytoplasmic ATPase complex is absent.

**Pseudorevertants of the $J_{(\Delta 13-24)}$ H* ΔN cells.** The *flhA(D456V)* or *flhA(T490M)* mutation partially restores motility to the ΔHI B* strain lacking FlgN[35] (Supplementary Fig. 7). Therefore, we introduced these alleles into the $J_{(\Delta 13-24)}$ H* ΔN cells by P22-mediated transduction to see whether motility was rescued. As expected, either of these *flhA* mutations improved motility and flagellar protein export (Fig. 3d and e), indicating that these *flhA* mutations can also activate the Na$^+$-driven export engine in the absence of FlgN. Therefore, we suggest that these single-residue substitutions in FlhA$_C$ allow FlhA$_C$ to adopt a conformation mimicking a FlgN-bound state of FlhA$_C$, thereby allowing the export gate complex to utilize the SMF to transport flagellar structural subunits to the cell exterior even in the absence of FlgN.

**Effect of deletion of FliJ residues 13–24 on the interactions of FlgN and FlhA$_C$.** The interaction of FliJ with the linker region of FlhA$_C$ (FlhA$_L$) is required for activation of the transmembrane export gate complex, and FliH and FliI are required for efficient interaction between FliJ and FlhA$_L$[21,32]. In addition to binding with high affinity to FlhA$_C$, FlgN binds to FliJ with a $K_D$ of 22 μM[39]. Thus, FlgN may be important in the docking of FliJ to FlhA$_C$, especially when the functions of FliH and FliI are compromised. Therefore, we first investigated whether the *fliJ* (Δ13–24) deletion mutation affects the interaction of FliJ with FlgN. FlgN co-purified with GST-FliJ(Δ13–24) (Fig. 5a), indicating that the deletion does not abolish interaction with FlgN. To assess the strength of the FliJ-FlgN interaction, we examined binding of FlgN to immobilized GST-FliJ or GST-FliJ(Δ13–24) by surface plasmon resonance (SPR). Steady-state analysis of the SPR data with a 1 to 1 binding model indicated that the $K_D$ values for the FliJ-FlgN and FliJ(Δ13–24)-FlgN interactions were 27.4 ± 13.0 μM and 7.74 ± 0.12 μM, respectively (Fig. 5b). Thus, the deletion actually increases the binding affinity by about 3.5-fold.

We next investigated whether FlgN participates in the interaction of FliJ(Δ13–24) with FlhA$_C$. The amount of FlhA$_C$ that co-purified with GST-FliJ(Δ13–24) was much lower than that with GST-FliJ (Fig. 5c and d, upper panels), in agreement with a previous report[21]. FlgN did not improve the interaction of FliJ (Δ13–24) with FlhA$_C$ (lower panels). These results suggest that a direct interaction between FlgN and FlhA$_C$ turns on Na$^+$-coupled protein export independently of FliJ when the FliJ-FlhA$_C$ interaction is compromised.

**Effect of the *flhA(D456V)* and *flhA(T490M)* mutations on Na$^+$-coupled protein export by cells lacking FliH, FliI, and FliJ.** FliJ is apparently more important for Na$^+$-coupled flagellar protein export than FliH and FliI[21] because the protein export activity of the ΔHIJ B* strain is much lower than that of the ΔHI B* strain (Fig. 1 and Supplementary Fig. 1). To confirm this, we first analyzed the number and length of flagellar filaments produced by the ΔHI B* and ΔHIJ B* cells (Fig. 4a and Supplementary Table 2). About 78.2% of the ΔHI B* cells produced the filaments with an average of 1.6 ± 0.7 filaments per cell ($n = 266$) and an average length of 7.8 ± 2.5 μm ($n = 50$) (Fig. 4c, d). In contrast, only 13.5% of the ΔHIJ B* cells produced the filaments with an average of 1.1 ± 0.2 filaments per cell ($n = 57$) and an average length of 5.1 ± 2.2 μm ($n = 50$) (Fig. 4c, d). Therefore, we conclude that FliJ is required for efficient activation of the export gate complex even in the absence of FliH and FliI.

Since FliJ is not critical for activation of the Na$^+$-driven export engine, we hypothesized that the interaction of FliJ with FlhA$_C$ may be required for efficient opening of the polypeptide channel for the substrate entry into the channel. Because it has been shown that the *flhA(D456V)* or *flhA(T490M)* mutation significantly increases the probability of entry of export substrates such as FlgD and FlgE into the polypeptide channel in the ΔHI B* cells[20], we investigated whether either of these two *flhA* mutations might overcome the effects of the loss of both FliJ and FlgN. To clarify this, we introduced these *flhA* alleles into the ΔHIJ B* strain and found that either mutation increased the secretion levels of FlgD and FliC by more than 10 fold (Fig. 4b). Consistently, more than 95% of ΔHIJ B* cells containing either *flhA(D456V)* or *flhA(T490M)* mutation had several flagellar filaments (Fig. 4a, c and Supplementary Table 2) although the average length of those filaments was almost the same as that of the much fewer filaments by the ΔHIJ B* strain (Fig. 4d). The loss of FlgN did not significantly reduce the secretion levels of FlgD and FliC by the ΔHIJ B* cells with either of the *flhA* mutation (Fig. 4b). Furthermore, of the ΔHIJ B* cells containing Δ*flgN*::*tetRA*, 20.8% with *flhA(D456V)* and 8.5% with *flhA(T490M)* cells produced a single flagellar filament (Fig. 4a) in a way similar to the ΔN mutant strain (Supplementary Fig. 6a). Therefore, we suggest that the *flhA(D456V)* and *flhA(T490M)* mutations can activate the export gate complex to become a highly efficient Na$^+$-driven engine in the absence of FliH, FliI, FliJ, and FlgN.

**Discussion**

The flagellar protein export machinery maintains its activity despite various internal and external perturbations. To do so, this export machinery has evolved to become a dual-fuel export machine to exploit both H$^+$ and Na$^+$ as the coupling ion[10]. The wild-type export engine predominantly uses H$^+$ as a coupling ion[21,23]. However, when the ATPase complex does not work properly, the export engine uses its Na$^+$ channel to continue flagellar assembly[10], but the mechanism of the switching of the coupling ion was unknown. Here, we show that an impaired interaction between FliJ and FlhA$_C$ caused by diminished ATPase activity activates Na$^+$-coupled protein export (Figs. 1 and 2). We also found that an interaction between FlgN, an export chaperone specific for FlgK and FlgL[33], and FlhA$_C$ becomes essential for Na$^+$-coupled protein export (Figs. 3 and 4). FlgN promotes the docking of FlgK and FlgL to the FlhA$_C$ platform of the export gate complex to facilitate rapid and efficient export of these proteins[35-37]. Therefore, the loss of FlgN reduces the secretion levels of FlgK and FlgL, resulting in a considerable reduction in the probability of filament formation at the tip of the HBB[38]. We found here that in the $J_{(\Delta 13-24)}$ H* cells, deletion of FlgN inhibits the export of FlgD and FlgE (Fig. 3b). This result suggests that FlgN acts not only as a substrate-specific export chaperone but also as a switch to activate a backup mechanism that in the absence of the FliHIJ ATPase complex, turns on the Na$^+$-driven

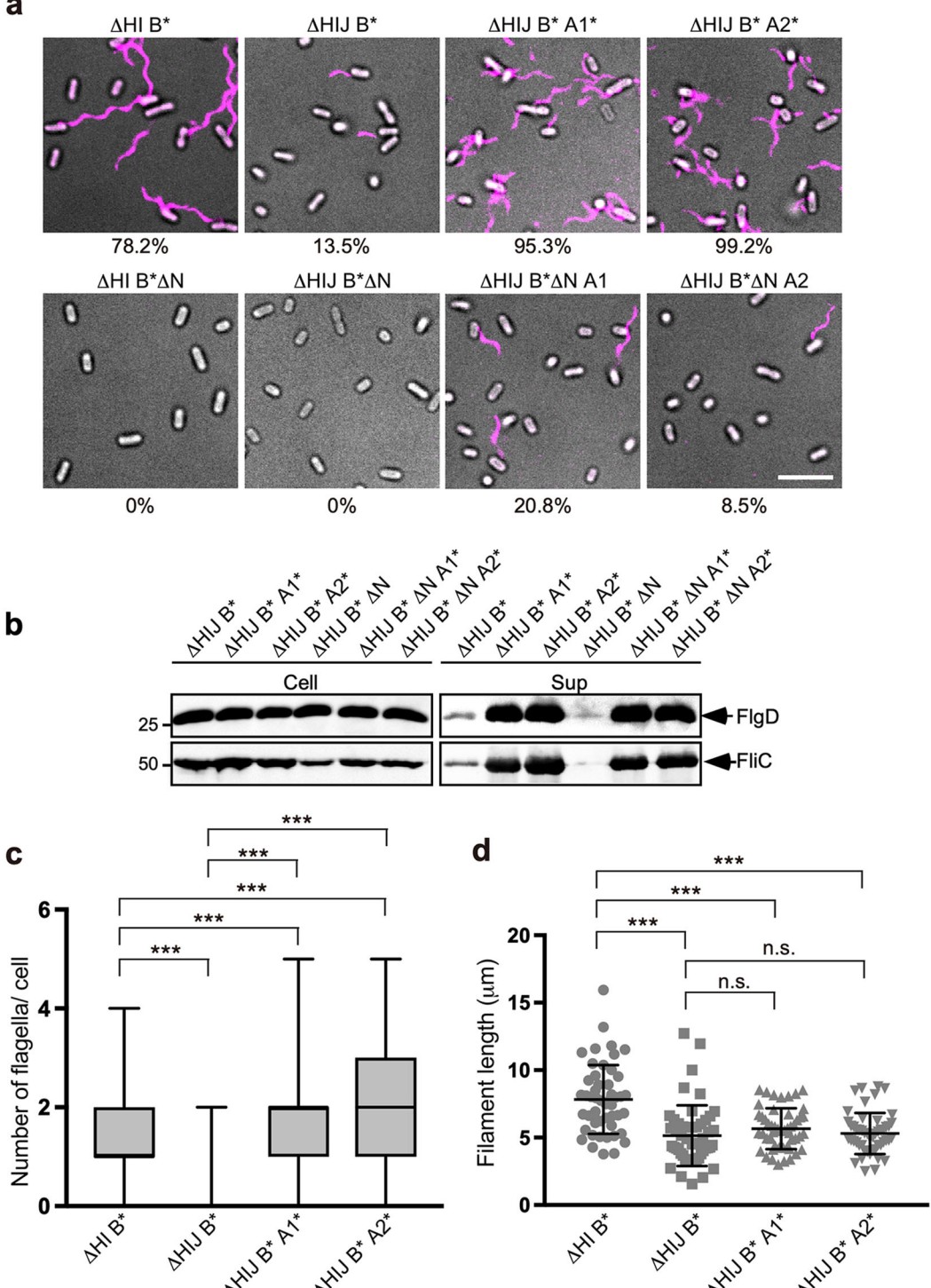

**Fig. 4 Effects of gain-of-function mutations in FlhA and deletion of FlgN on flagellar protein export and assembly by ΔHIJ B\* cells. a** Fluorescent images of MMHI0117 (ΔHI B\*), MMHIJ0117 (ΔHIJ B\*), MMHIJ0117-2 (ΔHIJ B\* A1\*), MMHIJ0117-3 (ΔHIJ B\* A2\*), MM9002 (ΔHI B\* ΔN), MM9004 (ΔHIJ B\* ΔN), MM9004-2 (ΔHIJ B\* ΔN A1\*), and MM9004-3 (ΔHIJ B\* ΔN A2\*). The cells were grown in T-broth (pH 7.5) containing 100 mM NaCl until the cells reached the stationary phase. Flagellar filaments were labelled with Alexa Fluor 594. The fluorescence images of the filaments labelled with Alexa Fluor 594 (magenta) were merged with the bright field images of the cell bodies. Scale bar, 5.0 μm. **b** Immunoblotting, using polyclonal anti-FlgD (1st row) or anti-FliC (2nd row) antibody, of whole-cell proteins (Cell) and culture supernatant fractions (Sup) prepared from the MMHI0117, MMHIJ0117, MMHIJ0117-2, and MMHIJ0117-3 cells. The regions of interest were cropped from original immunoblots shown in Supplementary Fig. 12. **c, d** Average number and length of flagellar filaments in the MMHI0117, MMHIJ0117, MMHIJ0117-2, and MMHIJ0117-3 cells. Filament length is the average of 50 filaments, and vertical lines are standard deviations. Comparisons between datasets were performed using a two-tailed Student's *t*-test. A *P*-value of <0.05 was considered to be statistically significant difference. \*\*\**P* < 0.001; n.s. no statistical significance. (See Supplementary Table 2).

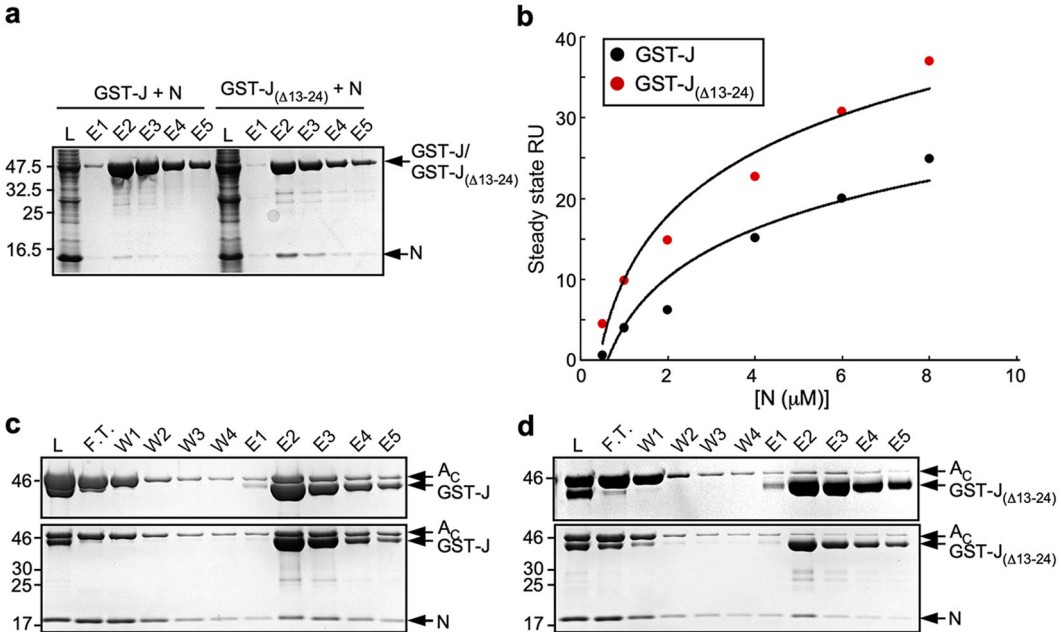

**Fig. 5 Effect of a deletion of residues 13–24 of FliJ on interactions of FliJ with FlgN and FlhA$_C$. a** Interaction between FliJ and FlgN. Cell lysates prepared from *Salmonella* SJW1368 (Δ*flhDC-cheW*) cells expressing either GST-FliJ (indicated as GST-J) or GST-FliJ(Δ13–24) (indicated as GST-J$_{(Δ13-24)}$) were mixed with those from *E. coli* BL21(DE3) Star cells producing His-FlgN (indicated as N), and then each mixture (indicates as L) was loaded onto a GST column. After extensive washing, proteins were eluted with a buffer containing 10 mM reduced glutathione. Eluted fractions were analyzed by SDS-PAGE with CBB staining. The regions of interest were cropped from original CBB-stained gels shown in Supplementary Fig. 13a. **b** Measurements of the binding affinities of FliJ and FliJ(Δ13–24) for FlgN by SPR. His-FlgN of various concentrations was flowed over the sensor surface with immobilized GST-FliJ or GST-FliJ(Δ13–24) in 10 mM HEPES pH 7.4, 0.15 M NaCl, 3 mM EDTA, 0.005% Surfactant P20 at a flow rate of 20 μl min$^{-1}$. All experiments were performed at 25 °C. The steady-state resonance units (RU) were plotted against FlgN concentrations. **c, d** Effect of FlgN on the FliJ-FlhA interaction. Purified His-FlhA$_C$ was mixed with purified **c** GST-FliJ or **d** GST-FliJ(Δ13–24) in the absence (upper panel) and presence (lower panel) of purified His-FlgN, and dialyzed overnight against PBS. Each mixture (L) was loaded onto a GST column. After washing with 10 ml PBS at a flow rate of about 0.5 ml min$^{-1}$, proteins were eluted with 10 mM reduced glutathione. Flow through fraction (F.T.), wash fractions (W), and elution fractions (E) were analyzed by SDS-PAGE with CBB staining. The regions of interest were cropped from original CBB-stained gels shown in Supplementary Fig. 13b.

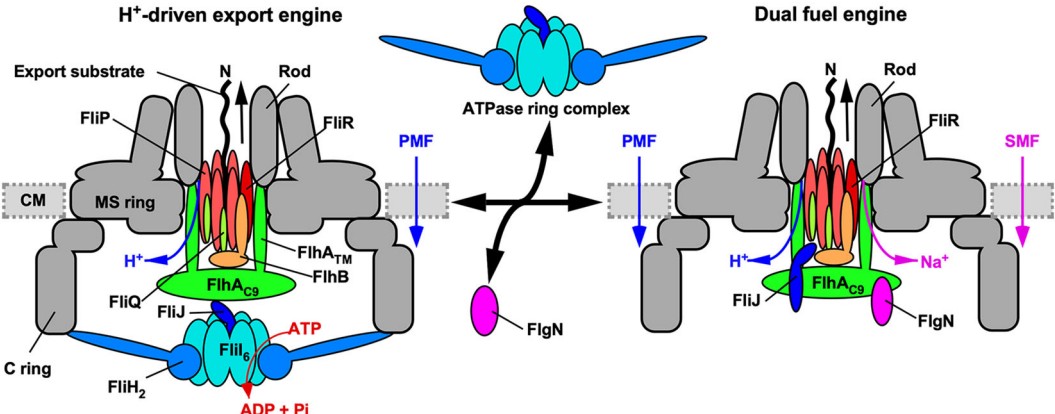

**Fig. 6 Schematic diagram of the flagellar protein export machinery.** The flagellar protein export machinery is composed of a transmembrane export gate complex made of FlhA, FlhB, FliP, FliQ, and FliR and a cytoplasmic ATPase complex consisting of FliH, FliI, and FliJ. The export gate complex is located inside the MS ring and utilizes proton motive force (PMF) across the cytoplasmic membrane (CM) to drive proton (H$^+$)-coupled flagellar protein export. FliP, FliQ and FliR form a polypeptide channel. FlhB associates with the FliP/FliQ/FliR complex and controls opening of the polypeptide channel. The C-terminal cytoplasmic domain of FlhA (FlhA$_C$) projects into the central cavity of the C ring. The N-terminal transmembrane domain (FlhA$_{TM}$) forms an ion channel for the translocation of H$^+$ and sodium ion (Na$^+$) from the periplasm to the cytoplasm. The cytoplasmic ATPase ring complex associates with the C ring through an interaction between FliH and a C ring protein, FliN. ATP hydrolysis by the FliI ATPase activates the export gate complex through an interaction between FliJ and FlhA$_L$ connecting FlhA$_C$ to FlhA$_{TM}$, becoming an active protein transporter to couple the H$^+$ flow through the FlhA channel to the translocation of export substrates into the polypeptide channel. When the cytoplasmic ATPase complex does not function properly, FlgN binds to FlhA$_C$ to open the Na$^+$ channel of FlhA$_{TM}$, allowing the export gate complex to utilize sodium motive force (SMF) across the cytoplasmic membrane to drive Na$^+$-coupled protein export.

export engine. FlgN interacts directly and with high affinity with FlhA$_C$ to accomplish this activation (Fig. 6).

The *flhA(D456V)* or *flhA(T490M)* mutation has been isolated as a bypass mutation of the motility defect of the ΔHI B* ΔN strain[35]. Here, we found that these *flhA* mutations overcome the effects of loss of both FlgN and FliJ (Fig. 4), suggesting that these two mutations allow FlhA$_C$ to adopt a conformation mimicking its conformation in the active FlgN/FliJ/FlhA$_C$ trimeric complex. Because the interaction between FliJ and FlhA$_C$ is not directly involved in activation of the Na$^+$-driven export engine, we propose that the interactions of FlhA$_C$ with FlgN and FliJ activate the Na$^+$ channel of FlhA and the polypeptide channel formed by FliP, FliQ, and FliR, respectively, so that the export gate complex efficiently couples the Na$^+$ flow through the FlhA channel with substrate entry into the polypeptide channel (Fig. 6).

FlgN binds to a well-conserved hydrophobic dimple of FlhA$_C$ formed by Asp-456, Phe-459, and Thr-490[35,36,40]. When the ATPase complex is functional, FliJ binds to the flexible linker region of FlhA (FlhA$_L$) connecting FlhA$_C$ to the N-terminal transmembrane domain that forms an ion channel[21,41]. This interaction activates the export gate complex to become an active H$^+$-driven export engine[21,42] (Fig. 6).This conclusion is supported by the observation that deletion of residues 328–351 of FlhA$_L$ significantly weakens the FliJ-FlhA$_C$ interaction (Supplementary Fig. 9a) but not the FlgN-FlhA$_C$ interaction (Supplementary Fig. 9b). FlgN bound to FliJ(Δ13–24) but did not restore the impaired interaction between FliJ(Δ13–24) and FlhA$_C$ (Fig. 5). FliJ not only binds to FlgN[39] but also to the FlgN/FlgK complex[35]. The FlgN/FlgK/FliJ trimeric complex docks to the FlhA$_C$ platform (Supplementary Fig. 9c)[35]. When the GST-FlgN/FlgK/FliJ complex was mixed with FlhA$_C$ lacking residues 328–351 of FlhA$_L$, only a very small amount of FliJ co-purified with this complex (Supplementary Fig. 9c), indicating that FliJ dissociates from FlgN upon binding of the FlgN/FlgK/FliJ complex to FlhA$_C$ lacking residues 328–351 of FlhA$_L$. Because protein transport activity was higher in the presence of FliJ than in its absence (Fig. 1), we propose that the cytoplasmic FlgN/FliJ complex docks to the FlhA$_C$ platform through an interaction between FlgN and FlhA$_C$, which then induces the dissociation of the FlgN/FliJ complex into FlgN and FliJ subunits to bind to the hydrophobic dimple of FlhA$_C$ and FlhA$_L$, respectively. These interactions then fully activate the Na$^+$-driven engine of the export gate complex in the absence of an active ATPase complex (Fig. 6). This conclusion is supported by the crystal structure of a FliJ homologue, CdsO, in complex with CdsV$_C$, which is a FlhA$_C$ homologue[43]. It remains unknown how FliJ binds to FlhA$_L$ because CdsO does not bind to the linker region of CdsV$_C$ in the crystal structure.

The 3′-5′ cyclic diguanylate monophosphate molecule binds to the FliI ATPase to inhibit the FliI ATPase activity[24]. This event might be expected to inhibit the H$^+$-coupled activity of the protein export channel. A subpopulation of planktonic cells is generated during biofilm development, perhaps as a "hedge-betting ploy" for cells to escape the biofilm[25]. Because the total PMF is quite low in the cells living in the biofilm structure[26], we propose that activation of the Na$^+$-driven export engine would provide a selective advantage for cells living in the biofilm.

## Methods

### Bacterial strains, plasmids, transductional crosses, and DNA manipulations.
Wild-type and mutant strains of *S. enterica* serovar Typhimurium and plasmids used in this study are listed in Supplementary Table 3. P22-mediated transductional crosses were carried out with p22HT*int*. DNA manipulations were performed using standard protocols. DNA sequencing reactions were carried out using BigDye v3.1 (Applied Biosystems) and then the reaction mixtures were analyzed by a 3130 Genetic Analyzer (Applied Biosystems).

### Motility assays in soft agar.
Fresh colonies were inoculated onto soft agar plates [1% (w/v) tryptone, 10 mM potassium phosphate pH 7.5, 0.35%(w/v) Bacto agar] or soft agar plates containing 100 mM NaCl or 100 mM KCl and incubated at 30 °C. At least ten independent measurements were performed. A diameter of the motility ring of each *Salmonella* strain was measured using ImageJ software version 1.52 (National Institutes of Health).

### Secretion assay.
Wild-type and mutant cells of *S. enterica* serovar Typhimurium were grown overnight in T-broth [1%(w/v) Bacto tryptone, 10 mM potassium phosphate pH 7.5] without 100 mM NaCl. A 50 μl of the overnight culture was inoculated into a 5 ml of fresh T-broth (pH 7.5) or T-broth (pH 7.5) containing 100 mM NaCl or 100 mM KCl and incubated at 30 °C with shaking until the cell density had reached an OD$_{600}$ of ca. 1.4–1.6. Cultures were centrifuged to obtain cell pellets and culture supernatants. The cell pellets were resuspended in a sample buffer solution [62.5 mM Tris-HCl, pH 6.8, 2% sodium dodecyl sulfate (SDS), 10% glycerol, 0.001% bromophenol blue] containing 1 μl of 2-mercaptoethanol. Proteins in the culture supernatants were precipitated by 10% trichloroacetic acid and suspended in a Tris/SDS loading buffer (one volume of 1 M Tris, nine volumes of 1× sample buffer solution)[44] containing 1 μl of 2-mercaptoethanol. After boiling proteins in both whole cellular and culture supernatant fractions at 95 °C for 3 min, these protein samples were separated by SDS–polyacrylamide gel (normally 12.5% acrylamide) electrophoresis (SDS-PAGE) and transferred to nitrocellulose membranes (Bio-Rad) using a transblotting apparatus (Hoefer). Then, immunoblotting with polyclonal anti-FlgD, anti-FlgE, anti-FlgK, or anti-FliC antibody was carried out using iBand Flex Western Device as described in the manufacturer's instructions (Thermo Fisher Scientific). Detection was performed with Amersham ECL Prime western blotting detection reagent (Cytiva). Chemiluminescence signals were captured by a Luminoimage analyzer LAS-3000 (GE Healthcare). All image data were processed with Photoshop software CS6 (Adobe). At least three independent experiments were performed.

### Observation of flagellar filaments with a fluorescent dye.
Wild-type and mutant cells of *S. enterica* serovar Typhimurium were grown at 30 °C in T-broth (pH 7.5) with or without 100 mM NaCl until the cells reached a stationary phase. The cells were attached to a cover slip (Matsunami glass, Japan), and unattached cells were washed away with motility buffer (10 mM potassium phosphate pH 7.0, 0.1 mM EDTA, 10 mM L-sodium lactate). A 1 μl aliquot of polyclonal anti-FliC serum was mixed with 50 μl of motility buffer and then 50 μl of the mixture was applied to the cells attached to the cover slip. After washing with the motility buffer, 1 μl of anti-rabbit IgG conjugated with Alexa Fluor 594 (Invitrogen) was added to 50 μl of motility medium, and then the mixture was applied. After washing with the motility buffer, the cells were observed by fluorescence microscopy[45]. Fluorescence images were analyzed using ImageJ software version 1.52 (National Institutes of Health).

### Preparation of HBBs.
Wild-type and mutant cells of *S. enterica* serovar Typhimurium were grown at 30 °C in 500 ml of L-broth until the cell density had reached an OD$_{600}$ of ca. 1.0. The cells were harvested by centrifugation (10,000 × g, 10 min, 4 °C) and suspended in 20 ml of ice-cold 0.1 M Tris-HCl pH 8.0, 0.5 M sucrose, followed by adding EDTA and lysozyme at final concentrations of 10 mM and 0.1 mg ml$^{-1}$, respectively. The cell suspensions were stirred for 30 min at 4 °C. Then, the cells were solubilized on ice for 1 h by adding Triton X-100 and MgSO$_4$ at final concentrations of 1%(w/v) and 10 mM, respectively. The cell lysates were adjusted to pH 10.5 with 5 M NaOH and centrifuged (10,000 × g, 20 min, 4 °C) to remove cell debris. After ultracentrifugation (45,000 × g, 60 min, 4 °C), pellets were resuspended in 10 mM Tris-HCl, pH 8.0, 5 mM EDTA, 1%(w/v) Triton X-100, and the solution was loaded a 20–50%(w/w) sucrose density gradient in 10 mM Tris-HCl, pH 8.0, 5 mM EDTA, 1%(w/v) Triton X-100. After ultracentrifugation (49,100 × g, 13 h, 4 °C), intact flagella, HBBs or MS-C rings were collected and ultracentrifuged (60,000 × g, 60 min, 4 °C). For intact flagella, pellets were resuspended in 50 mM glycine, pH 2.5, 0.1%(w/v) Triton X-100, and were incubated at room temperature for 30 min to depolymerize the filaments. After ultracentrifugation (60,000 × g, 60 min, 4 °C), pellets were resuspended in 50 μl of 10 mM Tris-HCl, pH 8.0, 5 mM EDTA, 0.1%(w/v) Triton X-100. Samples were applied to carbon-coated copper grids, followed by negative staining with 2%(w/v) uranyl acetate. Electron micrographs were recorded with a JEM-1011 transmission electron microscope (JEOL, Tokyo, Japan) operated at 100 kV and equipped with a F415 CCD camera (TVIPS, Gauting, Germany) at a magnification of ×5500, which corresponds to 2.75 nm per pixel.

### Pull-down assays by GST chromatography.
To identify the flagellar protein required for activation of the Na$^+$-driven export engine, GST-FliJ was over-produced in the *Salmonella* MMHI0117 strain, and then the cells were suspended in PBS (8 g of NaCl, 0.2 g of KCl, 3.63 g of Na$_2$HPO$_4$•12H$_2$O, 0.24 g of KH$_2$PO$_4$, pH 7.4 per liter) and sonicated. After centrifugation of cell lysates to remove undisrupted cells and insoluble membrane fractions, the soluble fractions were loaded onto a glutathione Sepharose 4B column (bed volume, 1 ml) pre-equilibrated with 20 ml of PBS. After extensive washing of the column with PBS, proteins were eluted with 50 mM Tris-HCl, pH 8.0, 10 mM reduced glutathione.

Fractions containing GST or GST-FliJ were identified by SDS-PAGE with Coomassie Brilliant blue (CBB) staining. Then, these fractions were analyzed by immunoblotting with polyclonal anti-FlhA$_C$, anti-FliM, anti-FlgN, or anti-FliT antibody.

To analyze the FlgN-FliJ interaction by GST affinity chromatography, cell lysates prepared from SJW1368 cells expressing GST-FliJ or GST-FliJ($\Delta$13–24), were mixed with those from the *Escherichia coli* BL21 (DE3) Star strain transformed with pMMGN130 (His-FlgN). To effect of deletion of residues 328–351 of FlhA$_L$ on the interactions of FlhA$_C$ with FlgN and FliJ, cell lysates prepared from SJW1368 expressing GST-FlgN was mixed with purified FlgK, purified His-FliJ and the soluble fraction isolated from *E. coli* BL21 (DE3) Star cells over-expressing either His-FlhA$_C$ or His-FlhA$_C$ lacking residues 328–351 of FlhA$_L$. Then each mixture was loaded onto a Glutathione Sepharose 4B column. After extensive wash of the column with PBS, bound proteins were eluted with 50 mM Tris-HCl, pH 8.0, 10 mM reduced glutathione.

His-FlhA$_C$, His-FlhA$_{C38K}$, and His-FlgN were overexpressed in *E. coli* BL21 (DE3) Star cells, and then these proteins were purified from the cell lysates by Ni affinity chromatography with a nickel-nitriloacetic acid (Ni-NTA) agarose column (QIAGEN). GST-FliJ, GST-FliJ($\Delta$13–24), and GST-FlgN were overexpressed in SJW1368 cells, and then these proteins were purified from cell lysates by GST affinity chromatography. To investigate the effect of deletion of residue 13–24 of FliJ on the FliJ-FlhA$_C$ interaction, purified His-FlhA$_C$ was mixed with purified GST-FliJ or GST-FliJ($\Delta$13–24) in the presence and absence of purified His-FlgN. To clarify the role of FlhA$_L$ on the interactions of FlhA$_C$ with FliJ and FlgN, purified His-FlhA$_C$ or His-FlhA$_{C38K}$ was mixed with purified GST-FliJ or GST-FlgN. Each mixture was dialyzed overnight against PBS at 4 °C with three changes of PBS. A 5 ml of each mixture was loaded onto a glutathione Sepharose 4B column and washed with 10 ml of PBS at a flow rate of ca. 0.5 ml min$^{-1}$. Bound proteins were eluted with 5 ml of 50 mM Tris-HCl, pH 8.0, 10 mM reduced glutathione. At least three independent experiments were carried out.

**Surface plasmon resonance (SPR)**. Anti-GST antibody was immobilized on a CM5 chip using a GST capture kit as described in the manufacturer's instructions (GE Healthcare). 40 μl of 10 μg ml$^{-1}$ GST-FliJ or 10 μg ml$^{-1}$ of GST-FliJ($\Delta$13–24) were injected over the chip pre-equilibrated with a binding buffer (10 mM HEPES pH 7.4, 0.15 M NaCl, 3 mM EDTA, 0.005% Surfactant P20) at a flow rate of 20 μl min$^{-1}$ and immobilized on the sensor chip via the anti-GST antibody. Forty microliter of His-FlgN of various concentrations in the binding buffer to monitor association was passed over the sensor surface and then washed with the buffer to monitor dissociation at a flow rate of 20 μl min$^{-1}$. An acidic buffer (10 mM Glycine-HCl, pH 2.2) was used for regeneration of the surface of the sensor chip by removal of the captured proteins and any associates. All experiments were done at 25 °C. To obtain the K$_D$ value, we analyzed SPR profiles using BIAevaluation software version 4.1 as described in the manufacturer's instructions (GE Healthcare). At least three independent SPR measurements were carried out.

**Statistics and reproducibility**. Statistical tests, sample size, and number of biological replicates are reported in the figure legends. Statistical analyses were done using Prism 7.0c software (GraphPad). Comparisons were performed using a two-tailed Student's *t*-test. A *P*-value of < 0.05 was considered to be statistically significant difference. *P < 0.05; **P < 0.01; ***P < 0.001.

**Reporting summary**. Further information on research design is available in the Nature Research Reporting Summary linked to this article.

## Data availability
All data generated during this study are included in this published article and its Supplementary Information files. Strains, plasmids, polyclonal antibodies and all other data are available from the corresponding author on reasonable request.

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

## Acknowledgements

We acknowledge Kelly T. Hughes for his kind gift of the *flgN::tetRA* allele, Kouhei Ohnishi for his kind gift of polyclonal anti-FlgM antibody, Yumi Inoue and Yasuyo Abe for technical assistance and Michael D. Manson for critical reading of the manuscript and helpful discussions. This work was supported in part by JSPS KAKENHI Grant Numbers JP26293097 and JP19H03182 (to T.M.), JP18K14638 and JP20K15749 (to M.K.), JP15H05593 and JP18H06159 (to Y.V.M.) and JP25000013 (to K.N.) and MEXT KAKENHI Grant Numbers JP15H01640 and JP20H05532 (to T.M.) and JP26115720 and JP15H01335 (to Y.V.M). This work has also been partially supported by JEOL YOKO-GUSHI Research Alliance Laboratories of Osaka University to K.N.

## Author contributions

T.M. and K.N. conceived and designed research; T.M., M.K., and Y.V.M. performed experiments; T.M., M.K., and Y.V.M. analyzed the data, and T.M. and K.N. wrote the paper based on discussion with other authors.

## Competing interests

The authors declare no competing interests.
