## [Peer Review File · Communications Biology]

Reviewers' comments:

Reviewer #1 (Remarks to the Author):

This builds on previous work from the same group, which made the surprising discovery that protein secretion by the flagellar type III secretion could use a sodium gradient in addition to a proton gradient to energize protein secretion when the FliH/IJ ATPase complex associated with secretion is not functioning properly.

The authors previously described a *fliJ* mutant deleted for residues 13-24 that was defective in interaction with FlhA, which is the ion conduit for coupling ion import to protein export. A two-residue deletion in FliH (96-97) suppressed this defect and restored some motility to the *fliJ* mutant. The authors speculated that the *fliJ* mutant effect on FlhA might be on its sodium channel function.

In the first chapter, the authors show that addition of 100 mM NaCl improved motility for three mutant strains: MM104-3 (*fliJ*(Δ 13-24) *fliH*(Δ 96-97)), MMHI0117 (Δ *fliHI* *flhB*(P28T)) and MMHIJ0117 (Δ *fliHIJ* *flhB*(P28T)). The main difference appears to be that MM104-3 is already motile and the presence of FliJ in MMHI0117 enhances motility compared to the FliH/IJ deletion in MMHI0117. Not surprisingly, enhanced motility in 100 mM NaCl correlates with enhanced secretion of substrates. Figure 1b shows that *fliC* is not expressed in HI0117 and HIJ0117 indicating that FlgM secretion is impaired. The effect of pH 7.5 is said to diminish the proton gradient. However, in the referenced article the effect of external pH on internal pH was with addition of sodium benzoate and at pH 7.5 the internal pH was the same with or without added benzoate (Figure 1 of referenced paper). I fail to find data to support the assertion that pH 7.5 results in significant reduction in the proton gradient.

The authors next show that cells have more and longer flagella with 100 mM NaCl. It is known that addition of NaCl turns on Spi1 including HilD which increases *flhDC* transcription and could easily account for more and longer flagella. What is the effect of 100 mM NaCl on *flhDC* transcription? The authors assume an SMF effect without considering other possibilities based on published work. The next chapter predicts that a flagellar protein is required for the export engine to prefer the SMF to the PMF. A GST-FliJ fusion inhibits FlhA-dependent export in the MMHI0117 strain. A GST-FliJ was able to co-purify with the FlgN protein and known secretion chaperone for FlgK and FlgL by preventing degradation of FlgK and FlgL in the cytoplasm. However, it was shown by a plasmid clone of the *flgKL* genes would alleviate the requirement for FlgN (see: Aldridge, et al. *Mol. Microbiol.* (2003) 49: 1333-45.). This interaction between FliJ and FlgN is good support that FlgN delivers FlgK and FlgL to the export apparatus through this interaction. It has been shown previously that FlgN interacts with FlhA. The authors want to test if FlgN is somehow involved in the SMF coupled to export. Deletion of *flgN* in the 104-3 strain resulted in no motility. Loss of FlgN was specific to FlgKL export as the authors observe no effect on early substrate export FlgD and FlgE. The authors need to look at FlgM export since a defect in FlgM export would inhibit *fliC* expression and would explain no filaments in Sup Fig 2A and no cellular FliC in Sup Fig 3b. The next chapter describes the isolation of pseudorevertants from the 104-3 *flgN* deletion strain. Two previously described alleles of *flhA* that suppressed *flgN* motility defect were tested on the 104-3 *flgN* deletion and as expected also suppressed in this background. The authors suspect the FlgN-FlhA interaction is needed for FlgD and FlgE secretion. That stream of logic is not apparent to this reviewer. Somehow, they conclude FlgN activates sodium driven export. I have no idea how that conclusion is come to – absolutely no idea. Not surprising another suppressor was a *flgM* null allele.

The next chapter looks at the effect of the *fliJ* 13-24 residue deletion on FlgN-FlhA interaction. They propose that FlgN might facilitate binding of FliJ to FlhA. The *fliJ* 13-24 residue deletion did not affect its ability to bind FlgN by pull-down, but SPR showed the K_d decreased about 3.5-fold indicating increased interaction of the deletion with FliJ. They show that FlgN did not improve interaction of FliJ with FlhA.

The next chapter reports to look at the effect of *flhA* alleles, which suppress the secretion defect in the absence of FlgN, on sodium-coupled export. There is nothing in this chapter that looks at sodium driven export. They show that FliJ has a positive effect on flagellar synthesis. This has been reported by others. Again, they suggest the sodium effect is export when they haven't ruled out other factors. I bet 200 mM KCl would give a similar result. That this is a salt concentration effect. Overall this manuscript reads as though the authors are trying to subjectively prove a sodium-export hypothesis, based on suppressor analysis of export-defective mutants and a positive effect

of 100 mM NaCl on secretion rather than objectively test this hypothesis and try to invalidate the sodium-dependent export hypothesis.

Line 65: add reference:

Erhardt, M., M.E. Mertens, F.D. Faniani, and K.T. Hughes. ATPase-independent type-III protein secretion in *Salmonella enterica*. *PLoS Genetics* (2014), 10: e1004800.

Page 5, first paragraph: Figure 1b shows that *fliC* is not expressed in HI0117 and HIJ0117 indicating that FlgM secretion is impaired. The authors should mention this.

Line 109: same thing. The authors state that transcription is not increased, but clearly *fliC* transcription must be increased with 100 mM NaCl. How else does one explain the difference in cellular FliC levels?

Guo et al. Showed that flagellar secretion increased with either NaCl or KCl:

Guo, S., I. Alshamy, K.T. Hughes, and F.F.V. Chevance. Analysis of factors that affect FlgM-dependent type III secretion for protein purification with *Salmonella Typhimurium*. *J. Bacteriol.* (2014), 196: 2333-2347.

Thus, the authors should repeat their results with 100 & 200 mM NaCl and 100 & 200 mM KCl based on the results of Guo et al..

Line 143: change "caused" to "resulted in"

Line 159-160: No, that is not what the results indicate. The results most likely indicate that FlgM secretion is defective.

Sentence lines 227-229. There is no evidence that FlgN turns on sodium-coupled export.

Reviewer #2 (Remarks to the Author):

This work by Minamino et al. examines how the flagellar export apparatus of *Salmonella enterica* can be driven by the sodium motive force SMF when the coupling to the proton motive force is somehow disrupted. The message is not nearly as easily extracted from the manuscript as my simple description might suggest.

First, I think the authors should explain in the Introduction, and mention in the Abstract, conditions under which having flagellar protein export driven by the SMF might be relevant. They leave any mention of this until the last paragraph of the discussion, in which they propose that within biofilms cells may have a lower proton motive force and profit from having another way to construct flagella that does not depend on the PMF. One then wonders why they did not look at flagellar synthesis under conditions in which the PMF is depleted and see whether the SMF becomes important under those circumstances.

Instead, they utilize a *fliJD13-34 fliHD96-97* mutant (MM104-3) to present evidence that the SMF is responsible for energizing the export of flagellar proteins in this strain. That is all right, as the mutant is no doubt easier to manipulate than the PMF. They also show that deletion of the *flgN* gene compromises the ability of the mutant to use the SMF and that FlgN physically interacts with FliJ and FlhA.

Here is what I understand of the central conclusion. In the wild type, FliJ interacts with FlhA to activate it as a proton channel. In the double mutant, the altered FliJ protein can still interact with FlhA, but primarily to activate it as a sodium ion channel. This activation relies on the FlgN protein in a second function in addition to its role as a chaperone for the export of FlgK and FlgL. FlgN is shown to interact with both FliJ and FlhA, data that support its role in mediating the activation of FlhA as a sodium channel. (The information may be in the text somewhere, but I was not clear whether FlgN is important for the activation of the proton channel activity of FlhA in the wild-type strain.)

It seems that this is the important information. However, the Results section goes on with an additional mutant strain (MMHI0117) and its derivative MMHIJ0117 and pseudorevertants that arise through mutations in *flhA* or by elimination of *FlgM*, the negative transcriptional regulator of *FliA*, the flagella-specific sigma factor. All of this became totally confusing to me so that I lost sight of what the real purpose of the study was. Fortunately, things calmed down a bit and became more clear in the Discussion.

I strongly suggest that the authors take stock of what it is that they really want the reader to understand. The Results section is far too much of a data dump, and I am not convinced that much of that data helps make their main point. Rather, I think it obscures it. The Abstract seems straightforward, and the data results presented in the paper should all be relevant to providing evidence for the conclusion that "the interaction of *FlgN* with *FlhA* opens the Na^+ channel in the export engine, thereby maintaining the protein export activity in the absence of the active ATPase complex." A sentence might be added to the Abstract to consider when this sort of activation would be useful, as per the last paragraph in the Discussion.

Reviewer #3 (Remarks to the Author):

Minamino et al. demonstrate extensive evidence for the mechanism by which sodium-powered flagellar export is activated. This is of great interest in the flagellar community, but also more widely, as an example of built-in redundancy in biology as well as dual-powered protein machinery.

The methods are a nicely blended mix of microbiology, biochemistry and microscopy, and the work is highly suitable for publication in *Communications Biology*. I enjoyed learning about aspects of the export apparatus and the level of detail that had gone into the experimental design to establish interactions between the components of the system.

I have the following minor comments:

P5L23: Why, in general, was 100 mM NaCl chosen as the sodium concentration of choice? It is OK, but I was just wondering why; much previous SMF work in the BFM has been done at 85 mM NaCl. In particular for Fig. 1 I was wondering if an extra plate measurement at, eg, 20 mM or 5mM NaCl could be done to show some trend in swim plate size vs sodium. This wouldn't be such a hard experiment to do and it might give us some idea if it was an 'all or nothing' approach, or if there was some tunability in the system with SMF.

P6L5: I was wondering in general in Fig. 2 also what the timing of the experiment was, in that surely filament length depends somewhat on cell cycle (time after division). We should be sampling a distribution of this, but could some comment be made here regarding this? You have measured the WT distribution of filament length and number, and perhaps this is somewhat correlated with cell shape and size, and life cycle? For example, I am unsure if published yet but there is some info of relevance in this preprint (and could be discussed from this manuscript's authors' previous work also) (<https://doi.org/10.1101/767319>).

P6L21: I am unfamiliar with the use of the phrase 'bait' here, perhaps that is my ignorance but not sure if it is totally clear, can we clarify?

P7L5 typo: mediated

P14L10-17: The flow here seems a little surprising, and it is the last paragraph? Why are we suddenly talking about biofilms and subpopulations? I felt that I needed a couple more sentences to transition from the last paragraph here, it just caught me a bit by surprise. Perhaps add some more info here to contextualise?

P15L10: for motility plates 7 independent measurements were done. Is this data included somewhere? Might be nice to plot mean/SD or scatter plot of all measurements to see variation/reproducibility of these motility plates, as an SI Figure.

P23Fig1, as above, can you comment on dependence on Na, or include a midpoint Na measurement for the plates.

Also, I appreciate the strain names are what they are because you made them but it is a little hard to follow at times, because we have 104-3, HI0117 and HIJ0117, but I found myself having to

constantly refer to caption or text to remember what was what. Is it perhaps possible to include phenotype on the figure somehow? I realise this is subtle but it does slow the reading down so if there is a way to define a phenotype descriptor throughout would help flow.

P24Fig.2: For panel C it would be better I think if this data was presented as a scatter plot with overlay Mean + SD, because we aren't able to tell easily from this what the distribution looks like, most relevantly if there are a small subpopulation of cells with very small filaments, or any difference between WT and 104-3 in terms of distribution. The number of measurements is high (Supp. Table 1/2), so it would be good to explore how this was distributed.

Fig3: This is a very nice figure, well put together, that combines EM with blot data. Well done.

SI Fig 2: Alexa594 is used both here and in Fig2, but in SI Fig 2 it is coloured red vs magenta in Fig. 2. Perhaps make consistent, I personally prefer red but it doesn't matter just a little odd to have two colours for same fluorophore.

Our responses are listed below.

To Reviewer #1:

This builds on previous work from the same group, which made the surprising discovery that protein secretion by the flagellar type III secretion could use a sodium gradient in addition to a proton gradient to energize protein secretion when the FliHIJ ATPase complex associated with secretion is not functioning properly.

The authors previously described a fliJ mutant deleted for residues 13-24 that was defective in interaction with FlhA, which is the ion conduit for coupling ion import to protein export. A two-residue deletion in FliH (96-97) suppressed this defect and restored some motility to the fliJ mutant. The authors speculated that the fliJ mutant effect on FlhA might be on its sodium channel function.

Re: Thank you so much for your comments.

In the first chapter, the authors show that addition of 100 mM NaCl improved motility for three mutant strains: MM104-3 (fliJ(Δ 13-24) fliH(Δ 96-97)), MMHI0117 (Δ fliHI flhB(P28T)) and MMHIJ0117 (Δ fliHIJ flhB(P28T)). The main difference appears to be that MM104-3 is already motile and the presence of FliJ in MMHI0117 enhances motility compared to the FliHIJ deletion in MMHI0117. Not surprisingly, enhanced motility in 100 mM NaCl correlates with enhanced secretion of substrates. Figure 1b shows that fliC is not expressed in HI0117 and HIJ0117 indicating that FlgM secretion is impaired.

Re: Both MMHI0117 and MMHIJ0117 strains did not secrete FlgM into the culture media in the absence of 100 mM NaCl. Addition of 100 mM KCl did not facilitate the secretion of FlgM by these two strains, either. Thus, only Na⁺ has a positive impact on FlgM secretion by these two strains (**Please see Supplementary Fig. 4 of the revised manuscript**).

The effect of pH 7.5 is said to diminish the proton gradient. However, in the referenced article the effect of external pH on internal pH was with addition of sodium benzoate and at pH 7.5 the internal pH was the same with or without added benzoate (Figure 1 of referenced paper). I fail to find data to support the assertion that pH 7.5 results in significant reduction in the proton gradient.

Re: Please see Supplementary Table S1 of the Supplemental Information of the referenced paper, which we extracted and attached below. The intracellular pH was

measured to be about 7.5 when external pH was 7.5. Thus, a chemical gradient of protons (ΔpH) is gone at an external pH value of 7.5. To confirm this, we measured intracellular pH of *Salmonella* cells using a pH sensor probe, pHluorin(M153R), and found that the intercellular pH was 7.41 ± 0.05 when the cells were exponentially grown in T-broth (pH 7.5). We now mentioned this point in text.

Supplementary Table 1. Effects of intracellular pH changes on flagellar motor rotation.

	Benzoate						
pH out		7.5	7.0	6.5	6.0	5.8	5.5
pH in	-	7.5	7.4	7.3	7.3	7.2	7.2
pH in	+	7.4	7.1	6.7	6.3	6.1	5.9

The authors next show that cells have more and longer flagella with 100 mM NaCl. It is known that addition of NaCl turns on Spi1 including HilD which increases flhDC transcription and could easily account for more and longer flagella. What is the effect of 100 mM NaCl on flhDC transcription? The authors assume an SMF effect without considering other possibilities based on published work.

Re: Thank you very much for your comments. Because it has been reported that increased ionic strength facilitates the export of a flagellum-specific anti-sigma factor, FlgM, by wild-type cells, enhancing motility in soft agar (Guo *et al. J. Bacteriol.* **196**, 2333–2347. 2014), we previously investigated whether Na^+ , Li^+ , K^+ or Mg^{2+} affects the levels of flagellar building blocks secreted by wild-type and $\Delta\text{fliH-fliI flhB}(P28T)$ cells (Please see Fig. 1A of Minamino *et al. PLOS Pathog.* **12**, e1005495, 2016, which is attached below). In wild-type cells, neither Na^+ , Li^+ , K^+ nor Mg^{2+} affected the secretion level of FlgD. In the $\Delta\text{fliH-fliI flhB}(P28T)$ mutant, Na^+ dramatically enhanced FlgD secretion whereas either Li^+ , K^+ or Mg^{2+} did not. Because Na^+ did not increase the cellular level of FlgD at all, this suggests that Na^+ is specific for this positive impact on flagellar protein export by the $\Delta\text{fliH-fliI flhB}(P28T)$ mutant.

Fig. 1A of the Minamino PLOS Pathog 2016 paper.

We also showed that the secretion level of FlgD by the $\Delta fliH$ - $fliI$ $flhB(P28T)$ mutant increased with an increase in external NaCl concentration and reached a maximum at 100 mM NaCl although their cellular levels were almost constant over a wide range of external NaCl concentration (**Please see Fig. 1B of Minamino *et al. PLOS Pathog.* 12, e1005495, 2016, which is also attached below).**

Fig. 1B of the Minamino PLOS Pathog 2016 paper.

Furthermore, we also showed that Na^+ facilitated the secretion of other flagellar building blocks such as FlgE, FliK, FlgK and FlgL by the $\Delta fliH$ - $fliI$ $flhB(P28T)$ mutant but did not affect cytoplasmic levels of those proteins (**Please see Fig. S4 of Minamino *et al. PLOS Pathog.* 12, e1005495, 2016, which is attached below).** Because we used SJW1103 as a wild-type strain for motility and chemotaxis studies, we assumed that Na^+ does not activate FlhDC transcription in the SJW1103 background.

Fig. S4 of the Minamino PLOS Pathog 2016 paper.

*The next chapter predicts that a flagellar protein is required for the export engine to prefer the SMF to the PMF. A GST-FliJ fusion inhibits FlhA-dependent export in the MMHI0117 strain. A GST-FliJ was able to co-purify with the FlgN protein and known secretion chaperone for FlgK and FlgL by preventing degradation of FlgK and FlgL in the cytoplasm. However, it was shown by a plasmid clone of the flgKL genes would alleviate the requirement for FlgN (see: Aldridge, *et al. Mol. Microbiol.* (2003) 49: 1333-45.). This interaction between FliJ and FlgN is good support that FlgN delivers FlgK and FlgL to the export apparatus through this interaction. It has been shown*

previously that FlgN interacts with FlhA. The authors want to test if FlgN is somehow involved in the SMF coupled to export. Deletion of flgN in the 104-3 strain resulted in no motility. Loss of FlgN was specific to FlgKL export as the authors observe no effect on early substrate export FlgD and FlgE. The authors need to look at FlgM export since a defect in FlgM export would inhibit fliC expression and would explain no filaments in Sup Fig 2A and no cellular FliC in Sup Fig 3b.

Re: Thank you so much for your comments. FlgN is known to act as a cytoplasmic chaperone specific for two hook-filament junction proteins, FlgK and FlgL, not only to protect them from degradation by proteases but also to support their efficient docking to FlhA. We cited the Aldridge Mol. Microbiol. 2003 paper and described this important information in the revised manuscript. In our present study, we found that removal of FlgN from the MM104-3 strain inhibited the secretion of FlgD and FlgE into the culture media **(Please see Fig. 3b of the revised manuscript)**.

Consistently, the MM104-3 $\Delta flgN$ strain produced only the MS-C ring structures **(Please see Fig. 3c of the revised manuscript)**. In contrast, removal of FlgN from the SJW1103 strain considerably reduced the secretion levels of FlgK and FlgL but did not affect the secretion levels of FlgD and FlgE at all **(Please see Fig. 3b of the revised manuscript)**. Consistently, the $\Delta flgN$ strain produced only hook-basal bodies in a way similar to the wild-type strain. Thus, FlgN exerts a positive impact on the export of all flagellar building blocks by the MM104-3 strain.

It has been reported that FlgM is secreted into the culture media upon completion of the hook structure, allowing FliA to induce the transcription of class III genes such as FlgK, FlgL and FliC **(Hughes et al. Science 1993; Kutsukake, MGG 1994)**. We analyzed the cellular levels of FlgK, FlgL and FliC by immunoblotting and found that deletion of FlgN results in a considerable decrease in the cellular levels of FlgK, FlgL and FliC, indicating that FlgN deletion inhibits the secretion of FlgM as well.

The next chapter describes the isolation of pseudorevertants from the 104-3 flgN deletion strain. Two previously described alleles of flhA that suppressed flgN motility defect were tested on the 104-3 flgN deletion and as expected also suppressed in this background. The authors suspect the FlgN-FlhA interaction is needed for FlgD and FlgE secretion. That stream of logic is not apparent to this reviewer. Somehow, they conclude FlgN activates sodium driven export. I have no idea how that conclusion is come to – absolutely no idea. Not surprising another suppressor was a flgM null allele.

Re: The FlgN deletion inhibited HBB assembly in the MM104-3 strain, thereby inhibiting the secretion of FlgD and FlgE into the culture media. However, the *flhA(D456A)* and *flhA(T490M)* mutations allowed the MM104-3 Δ *flgN* strain to secrete FlgD and FlgE to the culture media. Because FlgN can bind to FlhA_C even in the absence of its cognate substrates, FlgK and FlgL (Kinoshita *et al. Mol. Microbiol.* **2013**), this suggests that the interaction between FlgN and FlhA_C become important for the activation of the transmembrane export gate complex in the MM104-3 strain. To make our logic clear, we changed our description as follows:

“As expected, the *flhA(D456V)* and *flhA(T490M)* mutations improved motility of and flagellar protein export by the *fliJ(Δ13–24) fliH* ΔflgN::tetRA* cells (Fig. 3d and 3e), suggesting that the *flhA(D456V)* and *flhA(T490M)* mutations allow FlhA_C to adopt a certain conformation mimicking a FlgN-bound state of FlhA_C even in the absence of FlgN. Because FlgN alone binds to FlhA_C with nanomolar affinity³⁶, we suggest that the interaction between FlgN and FlhA_C activates the Na⁺-driven export engine of the transmembrane export gate complex in the *fliJ(Δ13–24) fliH strain, thereby allowing flagellar structural subunits to be transported to the cell exterior.”**

*The next chapter looks at the effect of the *fliJ* 13-24 residue deletion on FlgN-FlhA interaction. They propose that FlgN might facilitate binding of FliJ to FlhA. The *fliJ* 13-24 residue deletion did not affect its ability to bind FlgN by pull-down, but SPR showed the *K_d* decreased about 3.5-fold indicating increased interaction of the deletion with FliJ. They show that FlgN did not improve interaction of FliJ with FlhA.*

Re: Thank you so much for your understanding.

*The next chapter reports to look at the effect of *flhA* alleles, which suppress the secretion defect in the absence of FlgN, on sodium-coupled export. There is nothing in this chapter that looks at sodium driven export. They show that FliJ has a positive effect of flagellar synthesis. This has been reported by others. Again, they suggest the sodium effect is export when they haven't ruled out other factors. I bet 200 mM KCl would give a similar result. That this is a salt concentration effect.*

Re: As mentioned above, we previously showed that K⁺ did not promote the secretion of FlgD by the Δ *fliH-fliI flhB(P28T)* strain (Please see Fig. 1A of Minamino *et al. PLOS Pathog.* **12**, e1005495, 2016, which is attached above). We re-examined whether KCl facilitates motility of and flagellar protein export by the Δ *fliH-*

fliI flhB(P28T) and $\Delta fliH-fliI-fliJ flhB(P28T)$ strains in a way similar to NaCl and again found that K⁺ did not enhance motility of and flagellar protein export by these two strains (**Please see Supplementary Fig. 3 of the revised manuscript**).

Overall this manuscripts reads as though the authors are trying to subjectively prove a sodium-export hypothesis, based on suppressor analysis of export-defective mutants and a positive effect of 100 mM NaCl on secretion rather than objectively test this hypothesis and try to invalidate the sodium-dependent export hypothesis.

Re: Because we confirmed that KCl did not affect motility of and flagellar protein export by the $\Delta fliH-fliI flhB(P28T)$ and $\Delta fliH-fliI-fliJ flhB(P28T)$ mutants (**Please see Supplementary Fig. 3 of the revised manuscript**), we believe that the transmembrane export gate complex prefers to use Na⁺ as the coupling ion to drive flagellar protein export and assembly in the absence of the cytoplasmic ATPase complex and that the FlgN chaperone contributes to the efficient Na⁺-coupled protein export by the export gate complex.

Line 65: add reference:

Erhardt, M., M.E. Mertens, F.D. Faniani, and K.T. Hughes. ATPase-independent type-III protein secretion in Salmonella enterica. PLoS Genetics (2014), 10: e1004800.

Re: Cited this reference.

Page 5, first paragraph: Figure 1b shows that fliC is not expressed in HI0117 and HIJ0117 indicating that FlgM secretion is impaired. The authors should mention this.

Re: Agreed and mentioned in text.

Line 109: same thing. The authors state that transcription is not increased, but clearly fliC transcription must be increased with 100 mM NaCl. How else does one explain the difference in cellular FliC levels?

Re: We used P_{fliC}::luxCDABE to measure the *fliC* promoter activity in the presence and absence of 100 mM NaCl or KCl in the SJW1103 background. The *fliC* promoter activities were measured to be $1.36 \pm 0.33 \times 10^7$ RLU and $1.57 \pm 0.33 \times 10^7$ RLU in the presence of 100 mM KCl and 100 mM NaCl, respectively, whereas the promoter activity was $2.45 \pm 0.44 \times 10^7$ RLU in the absence of either KCl or NaCl. This is in agreement with our previous report showing that the transcription levels of flagellar

genes were not increased by adding 100 mM NaCl (**Minamino et al. PLOS Pathog. 12, e1005495, 2016**). Consistently, we found that neither Na⁺ nor K⁺ affected the steady cellular levels of flagellar proteins. However, we do not know why neither Na⁺ nor K⁺ exert the positive impact on flagellar gene transcription in the SJW1103 background. To avoid confusion, we would not like to show our transcription assay data until we will have a clear reason.

Guo et al. Showed that flagellar secretion increased with either NaCl or KCl: Guo, S., I. Alshamy, K.T. Hughes, and F.F.V. Chevance. Analysis of factors that affect FlgM-dependent type III secretion for protein purification with Salmonella Typhimurium. J. Bacteriol. (2014), 196: 2333-2347. Thus, the authors should repeat their results with 100 & 200 mM NaCl and 100 & 200 mM KCl based on the results of Gua et al.

Re: We carried out motility analysis and flagellar protein export assays as suggested by this reviewer and found that KCl did not facilitate motility of and flagellar protein export by the wild-type, MM104-3, $\Delta fliH-fliI flhB(P28T)$ and $\Delta fliH-fliI-fliJ flhB(P28T)$ strains (**Please see Supplementary Fig. 3 of the revised manuscript**). The motility ring size of wild-type cells was slightly larger in the presence of 100 mM NaCl and 100 mM KCl than in their absence presumably due to the slow growth rate in their absence because the motility ring size depends on the bacterial growth rate (**Please see Supplementary Fig. 2 of the revised manuscript**).

Line 143: change “caused” to “resulted in”

Re: Corrected

Line 159-160: No, that is not what the results indicate. The results most likely indicate that FlgM secretion is defective.

Re: Removal of FlgN from the MM104-3 strain inhibited the secretion of FlgD and FlgE into the culture media (**Please see Fig. 3b of the revised manuscript**). Consistently, the MM104-3 $\Delta flgN$ strain produced only the MS-C ring structures (**Please see Fig. 3c of the revised manuscript**). In contrast, the removal of FlgN from the SJW1103 strain considerably reduced the secretion levels of FlgK and FlgL but did not affect the secretion levels of FlgD and FlgE at all (**Please see Fig. 3b of the revised manuscript**). Consistently, the $\Delta flgN$ strain produced only hook-basal bodies in a way similar to the wild-type strain. These results indicate that FlgN becomes essential for the export of all flagellar structural subunits in the MM104-3

strain background rather than for FlgM secretion.

Sentence lines 227-229. There is no evidence that FlgN turns on sodium-coupled export.

Re: Pull-down assays by GST affinity chromatography revealed that FlgN did not improve the interaction between FliJ(Δ 13–24) and FlhA_C (**Please see Fig. 4 of the revised manuscript**). Unlike the FliS and FliT chaperones, which require their cognate substrates, FliC and FliD, respectively to bind to FlhA_C, FlgN alone can bind to FlhA_C (**Please see Fig. 4 of the revised manuscript**). Because FlgN is required for the activation of the Na⁺-driven export engine of the transmembrane export gate complex in the MM104-3 strain, we suppose that the FlgN-FlhA_C interaction turns on Na⁺-coupled protein export by the MM104-3 strain.

To Reviewer #2:

This work by Minamino et al. examines how the flagellar export apparatus of Salmonella enterica can be driven by the sodium motive force SMF when the coupling to the proton motive force is somehow disrupted. The message is not nearly as easily extracted from the manuscript as my simple description might suggest.

Re: Thank you so much for your useful comments and suggestions. We tried to make such message get clearer and easily extracted as much as possible.

First, I think the authors should explain in the Introduction, and mention in the Abstract, conditions under which having flagellar protein export driven by the SMF might be relevant. They leave any mention of this until the last paragraph of the discussion, in which they propose that within biofilms cells may have a lower proton motive force and profit from having another way to construct flagella that does not depend on the PMF. One then wonders why they did not look at flagellar synthesis under conditions in which the PMF is depleted and see whether the SMF becomes important under those circumstances.

Re: Thank you so much for your useful comments and suggestions. We described this point as suggested by this reviewer.

Instead, they utilize a fliJD13-34 fliHD96-97 mutant (MM104-3) to present evidence that the SMF is responsible for energizing the export of flagellar proteins in this strain. That is all right, as the mutant is no doubt easier to manipulate than the PMF.

They also show that deletion of the flgN gene compromises the ability of the mutant to use the SMF and that FlgN physically interacts with FliJ and FlhA.

Re: Thank you so much for your understanding.

Here is what I understand of the central conclusion. In the wild type, FliJ interacts with FlhA to activate it as a proton channel. In the double mutant, the altered FliJ protein can still interact with FlhA, but primarily to activate it as a sodium ion channel. This activation relies on the FlgN protein in a second function in addition to its role as a chaperone for the export of FlgK and FlgL. FlgN is shown to interact with both FliJ and FlhA, data that support its role in mediating the activation of FlhA as a sodium channel. (The information may be in the text somewhere, but I was not clear whether FlgN is important for the activation of the proton channel activity of FlhA in the wild-type strain.)

Re: Thank you so much for your understanding.

It seems that this is the important information. However, the Results section goes on with an additional mutant strain (MMHI0117) and its derivative MMHIJ0117 and pseudorevertants that arise through mutations in flhA or by elimination of FlgM, the negative transcriptional regulator of FliA, the flagella-specific sigma factor. All of this became totally confusing to me so that I lost sight of what the real purpose of the study was. Fortunately, things calmed down a bit and became more clear in the Discussion.

Re: Thank you so much for your comments. We needed to provide these mutant strains' data because these data fully support that the export gate complex becomes a Na⁺-coupled export engine when the cytoplasmic ATPase complex becomes non-functional and that the FlgN chaperone is required for the activation of the Na⁺-driven export engine. We agree, however, that the background information for the experimental design may have been rather limited for readers to fully understand the purpose of this study. We added necessary background information in the Abstract, Introduction and Results.

I strongly suggest that the authors take stock of what it is that they really want the reader to understand. The Results section is far too much of a data dump, and I am not convinced that much of that data helps make their main point. Rather, I think it obscures it. The Abstract seems straightforward, and the data results presented in the paper should all be relevant to providing evidence for the conclusion that "the

interaction of FlgN with FlhA opens the Na⁺ channel in the export engine, thereby maintaining the protein export activity in the absence of the active ATPase complex.” A sentence might be added to the Abstract to consider when this sort of activation would be useful, as per the last paragraph in the Discussion.

Re: Thank you so much for your comments. We rewrote the Results section to make our main point clearer as much as possible. We also rewrote the Abstract as follows:

“The bacterial flagellar protein export machinery is composed of a transmembrane export gate complex and a cytoplasmic ATPase complex. The export gate complex intrinsically has two distinct, H⁺- and Na⁺-driven export engines to facilitate the export of flagellar structural proteins. Planktonic *Salmonella* wild-type cells prefer to use the H⁺-driven export engine to construct flagella on the cell surface under a variety of environmental conditions. However, it remains unknown when and how the Na⁺-driven export engine is activated. To clarify this question, we analyzed the *fliJ*(Δ 13–24) *fliH*(Δ 96–97) mutant and provided evidence that the interaction of the FlgN chaperone with FlhA activates the Na⁺-driven export engine, thereby maintaining the protein export activity when the ATPase complex becomes non-functional. We propose that this type of activation would be useful for flagellar construction under conditions in which proton motive force across the cytoplasmic membrane is severely restricted.”

To Reviewer #3:

Minamino et al. demonstrate extensive evidence for the mechanism by which sodium-powered flagellar export is activated. This is of great interest in the flagellar community, but also more widely, as an example of built-in redundancy in biology as well as dual-powered protein machinery.

The methods are a nicely blended mix of microbiology, biochemistry and microscopy, and the work is highly suitable for publication in Communications Biology. I enjoyed learning about aspects of the export apparatus and the level of detail that had gone into the experimental design to establish interactions between the components of the system.

Re: Thank you very much for your supportive comments.

I have the following minor comments:

P5L23: Why, in general, was 100 mM NaCl chosen as the sodium concentration of choice? It is OK, but I was just wondering why; much previous SMF work in the BFM has been done at 85 mM NaCl. In particular for Fig. 1 I was wondering if an extra plate measurement at, eg, 20 mM or 5mM NaCl could be done to show some trend in swim plate size vs sodium. This wouldn't be such a hard experiment to do and it might give us some idea if it was an 'all or nothing' approach, or if there was some tunability in the system with SMF.

Re: We previously measured the levels of FlgD secreted by the $\Delta fliH-fliI flhB(P28T)$ mutant cells grown in T-broth (pH 7.5) containing 10, 25, 50 and 100 mM NaCl and showed that FlgD secretion levels increased with increasing external Na^+ concentration and reached a maximum at a Na^+ concentration of 100 mM (**Please see Fig. 1B of Minamino et al. PLOS Pathog. 12, e1005495, 2016, attached above**). Therefore, we analyzed the motility in the presence and absence of 100 mM NaCl.

P6L5: I was wondering in general in Fig. 2 also what the timing of the experiment was, in that surely filament length depends somewhat on cell cycle (time after division). We should be sampling a distribution of this, but could some comment be made here regarding this? You have measured the WT distribution of filament length and number, and perhaps this is somewhat correlated with cell shape and size, and life cycle? For example, I am unsure if published yet but there is some info of relevance in this preprint (and could be discussed from this manuscript's authors' previous work also) (<https://doi.org/10.1101/767319>).

Re: Thank you so much for your comments. The number of flagellar filaments and filament length were measured when the cells reached the stationary growth phase. We described this point in the figure legend.

P6L21: I am unfamiliar with the use of the phrase 'bait' here, perhaps that is my ignorance but not sure if it is totally clear, can we clarify?

Re: When searching for the target protein from a huge number of cellular proteins, the protein with a specific tag become a “bait” to catch its binding partner(s), just like fishing. Therefore, “bait” is often used in Biochemistry.

P7L5 typo: mediated

Re: Corrected

P14L10-17: The flow here seems a little surprising, and it is the last paragraph? Why are we suddenly talking about biofilms and subpopulations? I felt that I needed a couple more sentences to transition from the last paragraph here, it just caught me a bit by surprise. Perhaps add some more info here to contextualise?

Re: Thank you so much for your comments. We added the following sentences before start talking about biofilms.

“In *Salmonella* planktonic wild-type cells, the transmembrane export gate complex prefers to use PMF to transport flagellar structural subunits to the cell exterior^{16–18,21}. However, when the cytoplasmic ATPase ring complex becomes non-functional²⁴, which occurs in the biofilm formation, the export gate complex prefers to use SMF over a wide range of external pH¹⁰.”

P15L10: for motility plates 7 independent measurements were done. Is this data included somewhere? Might be nice to plot mean/SD or scatter plot of all measurements to see variation/reproducibility of these motility plates, as an SI Figure.

Re: We measured a diameter of the motility ring of 10 colonies of each strain in the presence and absence of 100 mM NaCl and provided relative diameter of each motility ring in Fig. 1a. The average diameter of the motility ring of each strain grown in the presence of 100 mM NaCl was set to 1.0.

P23Fig1, as above, can you comment on dependence on Na, or include a midpoint Na measurement for the plates.

Re: As mentioned above, we already showed that the secretion levels of flagellar building blocks increased with an increase in the external NaCl concentration and reached to a maximum at 100 mM NaCl (**Please see Fig. 1B of Minamino *et al.* PLOS Pathog. 12, e1005495, 2016, attached above**).

Also, I appreciate the strain names are what they are because you made them but it is a little hard to follow at times, because we have 104-3, HI0117 and HIJ0117, but I found myself having to constantly refer to caption or text to remember what was what. Is it perhaps possible to include phenotype on the figure somehow? I realise this is subtle but it does slow the reading down so if there is a way to define a

phenotype descriptor throughout would help flow.

Re: To make it easier to read the text, we changed “104-3”, “HI0117” and “HIJ0117” to “*fliJ*(Δ 13–24) *fliH**”, “ Δ *fliHI flhB**” and “ Δ *fliHIJ flhB**”, respectively.

P24Fig.2: For panel C it would be better I think if this data was presented as a scatter plot with overlay Mean + SD, because we aren't able to tell easily from this what the distribution looks like, most relevantly if there are a small subpopulation of cells with very small filaments, or any difference between WT and 104-3 in terms of distribution. The number of measurements is high (Supp. Table 1/2), so it would be good to explore how this was distributed.

Re: We changed this panel as suggested by this reviewer.

Fig3: This is a very nice figure, well put together, that combines EM with blot data. Well done.

Re: Thank you so much for this comment.

SI Fig 2: Alexa594 is used both here and in Fig2, but in SI Fig 2 it is coloured red vs magenta in Fig. 2. Perhaps make consistent, I personally prefer red but it doesn't matter just a little odd to have two colours for same fluorophore.

Re: Thank you for the point. The color of the flagellar filaments was unified to “magenta” for the readers with color blindness.

Reviewers' comments:

Reviewer #1 (Remarks to the Author):

My concerns have all been adequately addressed, nice work!

Reviewer #2 (Remarks to the Author):

I raised several points in my original review. One of them, leaving an explanation of the context in which a second, sodium-coupled export channel might be important for *Salmonella*, is now addressed in the Introduction. This research addresses an important problem and provides a convincing answer. The difficulty lies in the inaccessibility of the information to any but the most tenacious, diligent and experienced reader.

One point I raised in the original review was the confusion caused by all of the different strain names, which were repeated again and again as genotypes, which became very cumbersome and confusing. This point has not been dealt with adequately. I suggest that the authors include a table in which they assign each genotype they include in the results a simple name that they then use after the first description in the text. The reader will be able to consult the table to refresh memory. The important properties of each strain with respect to motility, flagellation pattern, and protein export capacity could also be summarized in this table or in an additional table or tables.

In the current manuscript, it is sometimes not clear which results are new and which are repeats from previous publications. It is annoying to read, "in agreement with previous results" followed by a literature citation, as it raises the question of what is really new. I understand that these are control experiments, but in the Results section they are often treated as though they are novel until that qualifying phrase appears.

The biggest problem is that the Results section is still almost unreadable. I will outline below what I think are the important findings, but I had to read and reread the manuscript and then synthesize for myself the major conclusions. Most readers will not be willing to do that. The authors should get the help of a native English speaking scientist who is familiar with bacterial flagellar assembly to assist with the production of a readable document. I am convinced that the data can be presented in a coherent fashion.

Here is my understanding of the results.

Motility of the *fliJ*(D13-24) *fliH*(D96-97) strain is poor because of a compromised interaction of *FliJ* of the *FliH**I**J* ATPase with *FliH**A**C*. This interaction is important for activating proton conduction through *FliH**A* to drive flagellar protein export. Motility is enhanced by 100 mM NaCl, leading to the conclusion that a sodium-coupled channel is important for export of flagellar proteins when the function of the ATPase complex is compromised. *FlgN*, the known chaperone for the capping proteins *FlgK* and *FlgL*, appears to be important for activation of this "bypass" sodium-coupled export. Unlike *FliJ*, which interacts with the linker region, *FliH**A**L*, that joins the transmembrane and cytoplasmic (*FliH**A**C*) domains of *FliH**A*, *FlgN* interacts directly with *FliH**A**C*. Two residue replacements, D456V and T490M, in the region of *FliH**A**C* that interacts with *FlgN* allow some sodium-coupled export of flagellar proteins in the absence of *FliH**I**J* and *FlgN*. This result suggests that interaction of *FlgN* with *FliH**A**C* causes a conformational change in *FliH**A* that converts it from a proton-conducting into a sodium-conducting channel. The D456V and T490M substitutions in *FliH**A**C* partially mimic the conformational change induced by *FlgN* binding to *FliH**A**C*. *FlgN*-mediated activation of sodium-coupled export of flagellar proteins may be important to allow for the presence of some motile cells under conditions of low proton motive force and high cyclic-di-GMP, as within biofilms. (There is some more involving the P28T substitution in *FliH**B* and the L244R substitution in *FliI* that I found hard to understand and integrate into the central story.)

If my understanding is correct, then it should be possible to present the data in a logical and digestible fashion. That is, in my opinion, what the authors need to do.

Reviewer #3 (Remarks to the Author):

Thank you for addressing my minor concerns. I recommend acceptance.

Our responses are listed below. We highlighted all changes in red in the revised manuscript (Marked Up version).

To Reviewer #1

My concerns have all been adequately addressed, nice work!

Re: Thank you so much for your support for publication of our manuscript in Communications Biology.

To Reviewer #2

I raised several points in my original review. One of them, leaving an explanation of the context in which a second, sodium-coupled export channel might be important for Salmonella, is now addressed in the Introduction. This research addresses an important problem and provides a convincing answer. The difficulty lies in the inaccessibility of the information to any but the most tenacious, diligent and experienced reader.

Re: Thank you so much for your useful comments.

One point I raised in the original review was the confusion caused by all of the different strain names, which were repeated again and again as genotypes, which became very cumbersome and confusing. This point has not been dealt with adequately. I suggest that the authors include a table in which they assign each genotype they include in the results a simple name that they then use after the first description in the text. The reader will be able to consult the table to refresh memory. The important properties of each strain with respect to motility, flagellation pattern, and protein export capacity could also be summarized in this table or in an additional table or tables.

Re: Thank you so much for your comments. We agreed with this reviewer and provided abbreviations for all strains which we used in this study and a table summarizing their important properties regarding motility, flagellar protein export and flagellar assembly.

In the current manuscript, it is sometimes not clear which results are new and which are repeats from previous publications. It is annoying to read, "in agreement with previous results" followed by a literature citation, as it raises the question of what is really new. I understand that these are control experiments, but in the Results section they are often treated as though they are novel until that qualifying phrase appears.

Re: Agreed and so we rewrote the Results section to make our new discoveries as clear as possible.

The biggest problem is that the Results section is still almost unreadable. I will outline below what I think are the important findings, but I had to read and reread the manuscript and then synthesize for myself the major conclusions. Most readers will not be willing to do that. The authors should get the help of a native English speaking scientist who is familiar with bacterial flagellar assembly to assist with the production

of a readable document. I am convinced that the data can be presented in a coherent fashion.

Re: Thank you so much for your comments. A native English speaking scientist helped us revise the manuscript to make the manuscript much more readable and impactful.

Here is my understanding of the results.

*Motility of the *fliJ*(D13-24) *fliH*(D96-97) strain is poor because of a compromised interaction of *FliJ* of the *FliHIJ* ATPase with *FlhAC*. This interaction is important for activating proton conduction through *FlhA* to drive flagellar protein export. Motility is enhanced by 100 mM NaCl, leading to the conclusion that a sodium-coupled channel is important for export of flagellar proteins when the function of the ATPase complex is compromised. *FlgN*, the known chaperone for the capping proteins *FlgK* and *FlgL*, appears to be important for activation of this “bypass” sodium-coupled export. Unlike *FliJ*, which interacts with the linker region, *FlhAL*, that joins the transmembrane and cytoplasmic (*FlhAC*) domains of *FlhA*, *FlgN* interacts directly with *FlhAC*. Two residue replacements, D456V and T490M, in the region of *FlhAC* that interacts with *FlgN* allow some sodium-coupled export of flagellar proteins in the absence of *FliHIJ* and *FlgN*. This result suggests that interaction of *FlgN* with *FlhAC* causes a conformational change in *FlhA* that converts it from a proton-conducting into a sodium-conducting channel. The D456V and T490M substitutions in *FlhAC* partially mimic the conformational change induced by *FlgN* binding to *FlhAC*. *FlgN*-mediated activation of sodium-coupled export of flagellar proteins may be important to allow for the presence of some motile cells under conditions of low proton motive force and high cyclic-di-GMP, as within biofilms. (There is some more involving the P28T substitution in *FlhB* and the L244R substitution in *FliI* that I found hard to understand and integrate into the central story.)*

Re: Thank you so much for your understanding. We removed a set of data regarding the L244R substitution in *FliI* and the *FlgM* deletion, both of which allow the *fliJ*(Δ 13–24) *fliH*(Δ 96–97) Δ *flgN* mutant strain to secrete *FlgD* and *FlgE* into the culture media in a Na⁺-dependent manner.

If my understanding is correct, then it should be possible to present the data in a logical and digestible fashion. That is, in my opinion, what the authors need to do.

Re: Yes, you are right. We tried to present our data in a logical fashion as much as possible.

To Reviewer #3

Thank you for addressing my minor concerns. I recommend acceptance.

Re: Thank you so much for your support for publication of our manuscript in Communications Biology.

REVIEWERS' COMMENTS:

Reviewer #2 (Remarks to the Author):

The authors have done an exemplary job of reformulating their manuscript and making it more accessible to readers. The use of simplified strain designations and the summary provided by Table 1 greatly strengthens the MS. I have no further reservations or suggestions for improvement.

Our responses are listed below.

To Reviewer #2

The authors have done an exemplary job of reformulating their manuscript and making it more accessible to readers. The use of simplified strain designations and the summary provided by Table 1 greatly strengthens the MS. I have no further reservations or suggestions for improvement.

Re: Thank you so much for your support for publication of our manuscript in Communications Biology. We really appreciate this reviewer for all helpful comments and suggestions for improving our manuscript.